

# LAKE 2.0: a model for temperature, methane, carbon dioxide and oxygen dynamics in lakes

Victor Stepanenko [1], Ivan Mammarella [2,3], Anne Ojala [4,3], Heli Miettinen [5], Vasily Lykosov [6,1], and Timo Vesala [2,3]

[1]Lomonosov Moscow State University, GSP-1, 119234, Leninskie Gory, 1, bld. 4, Moscow, Russia
[2]Department of Physics, P.O. Box 48, FI-00014, University of Helsinki, Finland
[3]Department of Forest Sciences, P.O. Box 27, FI-00014, University of Helsinki, Finland
[4]Department of Environmental Sciences, Niemenkatu 73, FI-15140 Lahti, University of Helsinki, Finland
[5]Department of Environmental Sciences, P.O. Box 65, FI-00014, University of Helsinki, Finland
[6]Institute of Numerical Mathematics, Russian Academy of Sciences, 119333, Gubkina str., 8, Moscow, Russia

*Correspondence to:* Victor Stepanenko (stepanen@srcc.msu.ru)

**Abstract.** A 1D model for enclosed basin (lake) is presented, reproducing temperature, horizontal velocities, oxygen, carbon dioxide and methane in the basin. All prognostic variables are treated in unified manner via generic 1D transport equation for horizontally averaged property. To the best of our knowledge, this is the most general form of 1D diffusion-type equation for a lake model available in the literature so far. Water body interacts with underlying sediments. These sediments are represented by a set of vertical columns with heat, moisture and methane transport inside. The model is validated vs. comprehensive observational dataset gathered at Kuivajärvi Lake (Southern Finland). Our results suggest that a gas transfer through thermocline under intense seiche motions is a bottleneck in quantifying greenhouse gas dynamics in dimictic lakes, calling for further research.

## 1 Introduction

Freshwater lakes occupy 1.3–1.8%, a comparatively small fraction of land surface globally (Downing et al., 2006). However, regional thermodynamic and dynamic effects of lakes on weather and climate are important for most of Canada, Finland, Western Siberia and some other regions (Dutra et al., 2010; Martynov et al., 2012; Eerola et al., 2014). This motivated inclusion of thermodynamic lake models into many Numerical Weather Prediction (NWP) and climate models (Martynov et al., 2012; Dutra et al., 2010; Mironov et al., 2010; Subin et al., 2012; Rontu et al., 2012).

The other mode of freshwater bodies impact on climate, is that through emissions of carbon dioxide ($CO_2$) and methane ($CH_4$) to the atmosphere (Tranvik et al., 2009). For instance, according to recent estimates (Bastviken et al., 2011), global methane flux from lakes offsets 25% of the esti-



mated land carbon sink, implying that lakes are an important component of global carbon cycle and climate system.

Concomitantly with growing awareness of lakes significance for current and future climate change, only few attempts have been made to develop lake models, incorporating thermodynamics, turbulence and biogeochemistry in order to simulate methane and carbon dioxide in natural water bodies

(Stepanenko et al., 2011; Kessler et al., 2012; Tan et al., 2015). The ultimate goal of these developments is to study the response of lakes and their greenhouse gas emissions under future climate change scenarios (Tan and Zhuang, 2015b) and through implementation of biogeochemical lake models into the Earth system models.

However, a number of problems arise concerning $CH_4$ and $CO_2$ modelling in lakes. First, va-

riety of biogeochemical processes involved in production and transformations of $CH_4$ and $CO_2$ are not well understood to an extent where rigorous mathematical description could be developed. For instance, methane production dependence on environmental factors has been tested in a bulk of studies (Borrel et al., 2011), however, to the best of our knowledge, only temperature dependence is quantified with high statistical confidence, e.g. (Yvon-Durocher et al., 2014). Moreover,

even a widely-accepted statement that methane is produced exclusively in anaerobic environment faces contradiction with some observational results (Damm et al., 2010), suggesting that there are $CH_4$ production mechanisms, not comprehended so far even at qualitative level. Second, lakes vary very much in climate, geological and biogeochemical environment resulting in enormous variability in greenhouse gas status (Juutinen et al., 2009). This situation is complicated by high vertical and

sometimes horizontal variability of gas concentrations in a given lake (Schilder et al., 2013; Blees et al., 2015). Third, when considering gas dynamics in lakes new physical processes become crucial, such as diffusion through the water surface (Donelan et al., 2002), vertical diffusion in metalimnion and hypolimnion, bubble interactions with sediments skeleton (Scandella et al., 2011), and others. Many of these have not been addressed enough so far in both theoretical and experimental studies.

The obstacles described above hinder development of mathematical model on fundamental process-understanding basis. Therefore, any lake greenhouse gas model would contain a number of empirical constants to be calibrated on an extensive dataset (Tan et al., 2015), what is a usual practice in e.g. wetland methane models (Walter et al., 1996; Walter and Heimann, 2000; Wania et al., 2009; Melton et al., 2013).

This work aims at developing the lake model based on rigorous mathematical development feasible in framework of 1D approach, applied for thermodynamic, hydrodynamic and biogeochemical prognostic variables in unified manner. We avoid using procedures for formal optimization (calibration) of the model parameters, rather focusing on qualitative behaviour of the model and its sensitivity to selected uncertain processes and constants. Vertical turbulent flux of dissolved gases through

hypolimnion and metalimnion are of special concern in this work. The lake model, developed here is based on LAKE model, that has been continuously advanced during last decade in Moscow State



University (Stepanenko and Lykossov, 2005; Stepanenko et al., 2011) and was extensively validated in LakeMIP experiments (Stepanenko et al., 2010, 2013, 2014) in terms of lake temperature and energy fluxes.

The model validation in terms of water temperature, $O_2$, $CO_2$ and $CH_4$ was performed using unique dataset collected by University of Helsinki at Kuivajärvi Lake, located near the SMEAR II station in Hyytiälä, Southern Finland (Hari and Kulmala, 2005).

The paper is organized as follows. Section 2 provides derivation of generic 1D equation that is applied then to temperature, horizontal velocities and dissolved gases. Section 3 introduces a reader

to a complex of measurements conducted at Kuivajärvi Lake and a model setup to simulate this lake. Then, Section 4 presents comparison of model results to observed data in a reference model run. In Section 5, we analyze results of reference experiment as well as of sensitivity experiments, elucidating significance of certain physical processes. Conclusions are summarized in Section 6.

## 2   The model overview

LAKE model is a one-dimensional model solving horizontally averaged equations for heat, gases and momentum transport for an enclosed water body. For taking into account heat and gases exchange with sloping bottom, the scheme for water temperature and gases concentrations is coupled to sediments columns originating at the bottom at different depths (see Section 2.5). Below we provide basics of 1D approach used and a general description for main groups of processes represented

in the model.

### 2.1   The generic 1D equation and vertical coordinate

We commence the description of LAKE model with derivation of a generic 1D lake modelling framework, implemented in current version of the model in respect to all prognostic variables. We confine ourselves to a concise summary of that derivation, while the interested reader will find a

rigorous mathematical development in Appendix A.

We start with the generic Reynolds-averaged advection-diffusion equation for the quantity $f$, that might be one of horizontal velocity components, temperature, turbulent kinetic energy (TKE), TKE dissipation or gas concentration (hereafter using summation over repeated indices):

$$c\frac{\partial f}{\partial t} = -\frac{\partial u_i f}{\partial x_i} - \frac{\partial F_i}{\partial x_i} + R_f(f,...), \tag{1}$$

assuming mass conservation equation for incompressible fluid:

$$\frac{\partial u_i}{\partial x_i} = 0, \tag{2}$$

where $u_i$ is the velocity component along $x_i$ Cartesian axis ($x_3 = z$ being an axis pointing along gravity and originating at a lake surface, $x_1 = x$, $x_2 = y$ - horizontal coordinates, $u_1 = u$, $u_2 =$





$v$, $u_3 = w$), $F_i$ is the sum of non-advective (turbulent and non-turbulent) fluxes of a property $f$ along

$x_i$, $c$ is an additional multiplier (specific heat in temperature equation, unity in other equations), and

$R_f$ standing for the sum of sources and sinks of $f$. Then, introduce the horizontal averaging operator

as:

$$\overline{f} = \frac{\int_{A(z)} f(x,y,z)dxdy}{A(z)}, \tag{3}$$

with $A(z)$ denoting the area of horizontal cross-section of a lake at depth $z$. After applying this

operator to (1) and making use of appropriate simplifications (Appendix A) we get:

$$c\frac{\partial \overline{f}}{\partial t} = \underbrace{-\frac{1}{A}\int_{\Gamma_{A(z)}} f(\mathbf{u_h} \cdot \mathbf{n})dl}_{\text{I.Advection by inlets, outlets and groundwater discharge}} \underbrace{+\frac{1}{A}\frac{\partial}{\partial z}\left(Ak_f\frac{\partial \overline{f}}{\partial z}\right)}_{\text{II.Turbulent diffusion/dissipation}}$$

$$\underbrace{-\frac{1}{A}\frac{\partial A\overline{F_{nz}}}{\partial z}}_{\text{III.Divergence of non-turbulent flux}} \underbrace{+\frac{1}{A}\frac{dA}{dz}(F_{nz,b}(z)+F_{tz,b}(z))}_{\text{IV.Contribution of the total vertical flux at the sloping bottom}}$$

$$\underbrace{+R_f(\overline{f},...)}_{\text{V.Horizontally averaged sum of sinks and sources}},$$

$$\tag{4}$$

where we have decomposed the total vertical flux $F_z = F_3$ into turbulent flux, $F_{tz}$, and a non-turbulent flux, $F_{nz}$, $F_z = F_{tz} + F_{nz}$; $\mathbf{u_h} = (u_1, u_2)$, $\mathbf{n}$ being an outer normal vector to the boundary $\Gamma_{A(z)}$ of cross-section $A(z)$, $k_f$ - turbulent diffusion/dissipation for variable $f$, and a subscript $b$ in-

dicating a variable's value at the sloping bottom. The vertical fluxes of quantity $f$ at lake's margins, $F_{tz,b}$, $F_{nz,b}$, we will thereafter call *marginal fluxes* for brevity (marginal heat flux, marginal gas flux, marginal friction, etc. For the horizontally mean turbulent flux we applied a first-order closure, $\overline{F_{tz}} = -k_f\frac{\partial \overline{f}}{\partial z}$. The non-turbulent fluxes enter equations for temperature (shortwave radiation flux) and for gases' concentrations (bubble flux).

In (4), we neglected terms containing vertical velocity, $w$. There are two of them. First is $\partial(\overline{w}\,\overline{f})/\partial z$, (Appendix A,(Omstedt, 2011) ) that is justified to omit for lakes with slow water level change during the simulation period considered. The second one is $\partial(\overline{w'f'})/\partial z$, $a' = a - \overline{a}$, $a = w, f$, representing the effect of vertical circulations of the scales larger than the Reynolds-averaging scale inherent to (1). The next paragraph considers the significance of this term.

The stratified enclosed water bodies under wind stress experience basin-scale circulations both above and below thermocline, the former induced by momentum flux from the atmosphere, and the latter - by pressure gradient caused by lake surface and thermocline tilt. Frequently these motions oscillate in time, known as surface (barotropic) and internal (baroclinic) seiches (Wüest and Lorke, 2003). Under Earth rotation, they transform to Kelvin and Poincare waves (Hutter et al., 2011). The





practice of 1D lake modelling, however, shows that under typical atmospheric forcing the top layer of
      a lake is almost always well-mixed (the so-called mixed layer or epilimnion) during ice-free period,
      so that any additional vertical mixing by basin-wide motion would not change vertical profiles there
      significantly. The well-mixed profiles below thermocline also may be produced involving simple
      seiche parameterization (Section 2.3), so that the explicit numerical treatment of closed vertical

circulation would not alter vertical distribution of water properties there as well. Situation changes
      when the thermocline tilt becomes significant, i.e. when the thin interface between epilimnion and
      hypolimnion reaches the lake surface at its margins (Shintani et al., 2010). In this case it is the term
      $\partial(\overline{w'f'})/\partial z$ that accounts for the eventual lake overturn, i.e. complete vertical homogenization of a
      water body. This process cannot be simulated by 1D lake models explicitly, but may be diagnosed

using Wedderburn (Shintani et al., 2010) and Lake numbers (Imberger and Patterson, 1989). Here,
      when applying the lake model for the lake under study, we will use Wedderburn number time series
      to check the validity of dropping out the "vertical circulation term" [1].

      Equation (4) is a generalization of equations that include lake shape effects encountered in many
      1D models designed for lakes (Stefan and Fang, 1994; Goudsmit, 2002; Jönhk et al., 2008; Tan and

Zhuang, 2015a) as well as for reservoirs (Zinoviev, 2014). In all 1D lake models we are aware of,
      term $IV$ does not include shortwave radiation flux in temperature equation and misses bubble flux
      of gases in equations for dissolved $CH_4$ and $CO_2$.

      The form of equation (4) written using geometric vertical coordinate $z$ is not convenient for the
      case of significant rate of water level change. In order to tackle this case, a normalized vertical

coordinate, $\xi = z/h(t)$, where $h$ is the maximal depth of a lake, has been introduced into equations
      of the model. Furthermore, the movement of $z$-axis origin when the surface level changes, strictly
      speaking, results in an additional term to a generic equation (4). Above leads to the following final
      form of (4):

$$c\frac{\partial \overline{f}}{\partial t} = -\frac{1}{A}\int_{\Gamma_{A(\xi)}} f(\mathbf{u_h} \cdot \mathbf{n})dl + \frac{1}{Ah^2}\frac{\partial}{\partial \xi}\left(Ak_f\frac{\partial \overline{f}}{\partial \xi}\right)$$

$$-\frac{1}{Ah}\frac{\partial A\overline{F_{nz}}}{\partial \xi} + \frac{1}{Ah}\frac{dA}{d\xi}[F_{nz,b}(\xi) + F_{tz,b}(\xi)] + R_f(\overline{f},...) + \left[\frac{\xi}{h}\frac{dh}{dt} - \frac{B_s}{h}\right]\frac{\partial \overline{f}}{\partial \xi} \qquad (5)$$

with $B_s$ signifying precipitation minus evaporation, i.e. the rate of $z$-axis origin motion, positive
      upwards. Although, it is the form (5) that is implemented in the LAKE model, it differs from (4) by
      metric terms only, so that for the sake of simplicity in subsequent flow we will refer to equation (4).
      Moreover, in this work we will keep lake depth $h$ constant, realistic for the lake under study.

      ---

      [1]Other possible mechanisms for basin-scale circulations include density currents along sloping bottom (Chubarenko,
      2010; Kirillin et al., 2015) during transitional seasons and ice period.



### 2.2 Lake thermodynamics

The water temperature in the model is driven by equation (4) with substitution $f \to T$, where $c = c_w \rho_{w0}$, $c_w$ - water specific heat, $\rho_{w0}$ - reference water density, $\overline{R_f} = 0$ (no heat sources in the water besides radiation heating), $\overline{F_{nz}(z)} = F_{nz,b}(z) = S_{rad}$ - shortwave radiation flux, positive downwards. The latter equality means that we assumed shortwave radiation flux to be horizontally homogeneous at all depths. This is commonly used approximation as getting data of the spatial distri-

bution of turbidity in a lake requires special measurements. Heat conductance is a sum of molecular and turbulent coefficients, $k_T = \lambda_m + \lambda_t$, where $\lambda_t = c_w \rho_{w0} \nu_T$ ($\nu_T$ – turbulent heat transfer coefficient, $\mathrm{m}^2/\mathrm{s}$) is computed from $k - \epsilon$ model (see Section 2.4).

Shortwave radiation flux, $S$, is treated as consisting of near infrared fraction and the rest energy (mostly visible radiation). Near infrared part is consumed completely at the surface, whereas the

155 visible fraction is partially reflected according to water albedo, and its remainder is attenuated with depth according to widely-used Beer-Lambert law with extinction coefficient specific for a lake under study (see Section 3.2).

To solve (4) for temperature one needs to specify top and bottom boundary conditions as well as a method for calculation of marginal heat flux, $F_{tz,b}(z)$, at each depth $z$. The top boundary condition

is a well-established heat balance equation, involving net radiation and a scheme for turbulent heat fluxes in a surface atmospheric layer based on Monin-Obukhov similarity theory (Paulson, 1970; Businger et al., 1971; Beljaars and Holtslag, 1991). The way of coupling the water column to bottom sediments through lower boundary condition and marginal heat flux is less straightforward. When the heat transfer in bottom sediments is solved by diffusion-lake equation, there are two options for

imposing boundary conditions at the "water-sediments" interface:

– Continuity of both heat flux and temperature at the interface;

– Continuity of heat flux across the interface and a method for heat flux calculation, relating it to in-water temperature gradient, e.g. through logarithmic profile formulae.

We found that the first option provides reasonable results for temperature and especially for gas

concentrations (see below in the paper), whereas the second one needs calibration of parameters entering the flux-gradient relationship in the bottom boundary layer. The marginal heat flux is calculated using the same temperature and flux continuity condition, that is facilitated by the solution of vertical heat transfer in sediments below sloping bottom (see details in Section 2.5).

The model also includes multilayer sediments, snow and ice modules (Stepanenko and Lykossov,

2005; Stepanenko et al., 2011) that are not used in this study.



### 2.3 Lake hydrodynamics

Applying the form (4) to horizontal momentum equations is straightforward with $F_{nz} = 0$, $c = 1$ and $R_f$ representing Coriolis force and horizontal pressure gradient. The Coriolis force has to be included in the momentum equations for lakes with horizontal size exceeding internal Rossby deformation radius (Patterson et al., 1984), that we will check below when validating the model for the lake under study.

The term $F_{tz,b}(z) A^{-1} dA/dz$ has the sense of *marginal friction* for the case of momentum equations. This term can be parameterized as quadratic in velocity with tunable proportionality coefficient (Jöhnk, 2001). Instead, we apply logarithmic layer friction with effective bottom roughness length, $z_{0b,eff}$. The characteristic "effective" in respect to $z_{0b,eff}$ accounts for the fact that while calculating bottom friction we use horizontally averaged velocity components $\overline{u}, \overline{v}$ instead of the velocity components' values in the logarithmic layer adjacent to bottom. As there are no theoretical hints how $z_{0b,eff}$ relates to the "true" bottom roughness, $z_{0b}$, it may be used as tunable parameter. However, our modelling results show, that choosing $z_{0b,eff}$ of the order of $z_{0b}$ expected at the bottom eventually provides reasonable results in terms of vertical mixing of water properties.

The more interesting story comes with parameterization of horizontal pressure gradient. We represent it at any depth $z$ as

$$-\frac{1}{\rho_{w0}} \frac{\partial p}{\partial x_i} = -g \frac{\partial h_s}{\partial x_i}, \; i = 1, 2, \tag{6}$$

($h_s$ - lake surface deviation from horizontal), implying that we have used hydrostatic equation with constant density, $\rho_{w0}$. This is a barotropic approximation since we neglected buoyancy in the hydrostatic equilibrium [2]. It is the simplest way to account for horizontal pressure differences still being capable to induce significant mixing below the thermocline (see below in the paper).

To estimate terms (6) in 1D model we modify the scheme proposed originally by U.Svensson (Svensson, 1978; Goudsmit, 2002). Fig. 1 provides a concept of the scheme. The parameterization takes the form:

---

[2]The only place in the model where buoyancy expressed by temperature fluctuations is taken into account is the $k - \epsilon$ closure, Section 2.4. It formally adds baroclinicity to the model, however, only in subgrid scale stress/fluxes.





$$g \overline{\frac{\partial h_s}{\partial x}} \approx \frac{g\pi^2}{4} \frac{h_{s,x2} - h_{s,x1}}{L_{x,0}}, \tag{7}$$

$$g \overline{\frac{\partial h_s}{\partial y}} \approx \frac{g\pi^2}{4} \frac{h_{s,y2} - h_{s,y1}}{L_{y,0}}, \tag{8}$$

$$\frac{dh_{s,y2}}{dt} = -\frac{dh_{s,y1}}{dt} = \frac{2}{A_0(t)} \int_0^1 v L_x h d\xi, \tag{9}$$

$$\frac{dh_{s,x2}}{dt} = -\frac{dh_{s,x1}}{dt} = \frac{2}{A_0(t)} \int_0^1 u L_y h d\xi, \tag{10}$$

where $L_x$, $L_y$ are the sizes of horizontal water body intersection in $x$ and $y$ directions, respectively, subscript "0" denotes values at the lake surface. For simplicity, in the model we approximate the lake's horizontal cross-section, $A(z)$ as an ellipse, so that $L_x$ stands for twice of major semiaxis and $L_y$ – for twice of minor semiaxis, or vice versa. Equations (9)-(10) express the change of surface level of four lake's sections ($h_{s,x1}$ being the mean of $h_s$ over the "left" section of a lake, $x < x_c$,

$h_{s,x2}$ – the same for the "right" lake section $x > x_c$, $h_{s,y1}$ – for $y < y_c$, $h_{s,y2}$ – for $y > y_c$, $(x_c, y_c)$ standing for the lake center) due to volume discharge through two vertical planes, $x = x_c$ and $y = y_c$, (Fig. 1), neglecting inflows and outflows. The multiplier $\pi^2/4$ in (7)-(8) arises instead of a "natural" choice of 2 in order the solution of the model equations for specific case of rectangular channel be matching the period of the 1-st surface seiche mode, i.e. the Merian formula (Merian, 1828) (see

Appendix C for mathematical development). According to that, the output of LAKE model for the case of 1D flow developing along a non-rotating channel after initial disturbance of lake surface demonstrates oscillations with a period very close to that predicted by Merian formula (not shown here). In the following, we will refer to 7-10 as either "surface seiche parameterization"or "dynamic pressure gradient parameterization".

Boundary conditions for momentum equations are momentum flux from the atmosphere, calculated according to air surface layer bulk formulae (Paulson, 1970; Businger et al., 1971; Beljaars and Holtslag, 1991), and friction at the deepest part of bottom following quadratic dependence on velocity with Chézy. Momentum flux accelerating currents is parameterized as a fraction of total momentum flux from the atmosphere (Stepanenko et al., 2014), because in conditions of limited fetch

(small lakes) a part of total momentum flux is consumed by wave development. Partitioning momentum flux between waves and in-water currents significantly reduces shear-driven vertical mixing during summertime.

### 2.4  Turbulence closure

The turbulence closure is a $k - \epsilon$ model with Canuto stability functions (Canuto et al., 2001). Non-

turbulent flux, $\overline{F_{nz}}$ in (4) is put to zero, because this model does not include any fluxes of $k$ and $\epsilon$ besides advection and turbulent transport. We neglect also advection of TKE and dissipation rate





by inlets and outlets (term $I$ in (4)), because there are no observation data or reasonable ways to theoretically estimate $k$ and $\epsilon$ in streams. Marginal flux is set as $F_{tz,b} = 0$ for TKE that is an exact boundary condition for logarithmic layer. For $\epsilon$, $F_{tz,b}$ is set to 0 as well, because non-zero flux

condition for TKE dissipation in the logarithmic layer (Burchard and Petersen, 1999) cannot be realized in this model framework (variables entering this condition are not available in the bottom boundary layer as they are averaged over horizontal). The top and bottom boundary conditions for TKE and dissipation equations are that for logarithmic layer (Burchard and Petersen, 1999). Sinks and sources of TKE and dissipation rate, i.e. buoyancy term and shear production, hidden in $\overline{R_f}$ of

(4), are approximated using only vertical derivatives of horizontally averaged temperature, salinity and velocity components. For constants of $k-\epsilon$ model used in this study, see Appendix B.

In our study the turbulence closure briefly described above will be referred to as "standard $k-\epsilon$ model". Pertinent to objectives of the study, we will also use extensions of standard $k-\epsilon$ model to account for specific mixing mechanisms in the thermocline, namely, gravity waves (Mellor, 1989)

and internal seiches (Goudsmit, 2002).

### 2.5 Heat and moisture processes in sediments

Snow and ice modulae are not used in this study. Processes in sediments are treated inside a set of 3D figures, that all have the same vertical dimension, $h_{sed}$, and the horizontal intersections of which are confined by sequential isobaths (Fig. 2). In each such a column all properties of sediments

are assumed to be horizontally homogeneous, so that only the vertical transport of heat and other quantities applies. Each column of sediments exchanges heat and methane with the horizontal water layer bounded from below and above by respective isobath levels according to continuity of flux and a quantity considered (temperature, methane concentration, see Section 2.2).

The heat processes in the model include vertical transport and phase transition between water

and ice. The vertical transport in sediments is described according to (Côté and Konrad, 2005). Liquid water is transported via gravity and capillary-sorption forces (Stepanenko and Lykossov, 2005). The latter are represented by diffusion-like term. Bottom boundary condition for temperature is geothermal heat flux, usually set to zero. For moisture, saturation of sediments is used for the top boundary and zero flux is applied at the bottom.

### 2.6 Biogeochemistry and transport of CH$_4$, CO$_2$ and O$_2$

The general scheme representing sources, sinks and transport mechanisms governing concentration of methane, carbon dioxide and oxygen (O$_2$) in the model is given in Fig 3.





### 2.6.1 Methane in sediments

Elaborate description of $CH_4$ model in sediments can be found in (Stepanenko et al., 2011), whereas
here we provide a general overview and latter amendments to the model presented therein. This
model is applied in every column of sediments under a lake.

In each column of sediments (Section 2.5) methane transport is considered to be vertical only. The
governing equation for bulk methane concentration, $C_{CH_4}$, reads:

$$\frac{\partial C_{CH_4,s}}{\partial t} = \frac{\partial}{\partial z} k_{CH_4,s} \frac{\partial C_{CH_4,s}}{\partial z_s} + P_{CH_4,s} - E_{CH_4,s} - O_{CH_4,s}, \tag{11}$$

where $k_{CH_4,s}$ designates molecular diffusivity of methane, $P_{CH_4,s}$ - production rate, $E_{CH_4,s}$ - ebul-
lition rate, $O_{CH_4,s}$ - aerobic oxidation rate (anaerobic oxidation is omitted), and $z_s$ denotes a vertical
coordinate originating at the column top. Vegetation uptake of methane by roots and aerenchyma
transport are neglected in this study. Methane production rate is confined to the upper part of a
sediments column and controlled by temperature by exponential dependence:

$$P_{CH_4,s} = P_0 \exp(-\alpha_{new} z_s) \mathrm{H}(T - T_{mp}) q_{10}^{T/10} (1 + \alpha_{O_2,inhib} C_{O_2,s})^{-1}, \tag{12}$$

with $P_0$ being a calibrated constant, reflecting the amount and quality of organic material in sedi-
ments in respect to methane production, $\alpha_{new} = 3\,\mathrm{m}^{-1}$ - a constant controlling the decrease of $CH_4$
production with depth, H is a step (Heaviside) function, $q_{10} = 2.3$ (Liikanen et al., 2002) - temper-
ature dependency constant, $T_{mp}$ - melting point temperature, $\alpha_{O_2,inhib}$ - a constant describing the
rate of inhibition of methane production with rise of bulk oxygen concentration in sediments, $C_{O_2,s}$.
The latter constant is set as $\alpha_{O_2,inhib} = 316.8\,\mathrm{m}^3/\mathrm{mol}$ to ensure 100 times inhibition of methane
production at oxygen content of 10 ppm, implying almost complete suppression of methanogenic
*Archaea* activity under this concentration (Borrel et al., 2011). Parameterization (12) traces back
to (Walter et al., 1996), the last multiplier added in this study. Deep methane production from old
organics (Stepanenko et al., 2011) at the bottom of talik is not included as the lake under study is not
a thermokarst one.

Ebullition rate, $E_{CH_4,s}$ becomes non-zero when bulk methane concentration exceeds a critical
value, defined by hydrostatic load of water column and sediments layer above at a given depth, $z_s$,
as well as by nitrogen concentration at the sediments top (Stepanenko et al., 2011; Walter et al.,
1996). Retention of bubbles in a sediments skeleton (Scandella et al., 2011) is neglected, so that
methane ebullition flux at the sediments top of $k$-th column, $F_{B,1,k}$, is calculated as

$$F_{B,1,k} = \int_0^{h_{sed}} E_{CH_4,s} dz_s \tag{13}$$



with $h_{sed}$ signifying the depth of sediments column.

Oxidation of methane in sediments takes place in the uppermost numerical layer only, where
oxygen concentration is assumed to deplete exponentially towards very small value at the base of
this layer. At the top, a continuity of oxygen concentration across the water-sediments interface is
applied. Then, a mean bulk oxygen concentration over the top numerical layer is calculated from
exponential law. Given bulk oxygen concentration, $C_{O_2,s}$, aerobic methane oxidation is calculated
according to Michaelis-Menthen kinetics

$$O_{CH_4,s} = V_{max,s} \frac{C_{CH_4,s}}{K_{CH_4,s} + C_{CH_4,s}} \frac{C_{O_2,s}}{K_{O_2,s} + C_{O_2,s}}, \tag{14}$$

where $V_{max,s} = 1.11*10^{-5} \, \mathrm{mol}/(\mathrm{m}^3 * \mathrm{s})$, $K_{CH_4,s} = 9.5*10^{-3} \, \mathrm{mol}/\mathrm{m}^3$, $K_{O_2,s} = 2.1*10^{-2} \, \mathrm{mol}/\mathrm{m}^3$
are methane oxidation potential and two half-saturation constants, respectively (Lidstrom and Somers,
1984).

In order the above scheme for methane oxidation and methane production inhibition to be realistic,
the top numerical layer in sediments is set to be of thickness typical for oxygenated layer in lake's
sediments, $1 \, \mathrm{cm}$ (Huttunen et al., 2006).

### 2.6.2 Methane in water

Methane concentration in water evolves according to equation of the form (4) with the term $I$
representing the input of $CH_4$ by inlets and its outflow by outlets (not taken into account in this
study). Diffusion coefficient, $k_{CH_4,w}$, is set equal to heat conductivity (turbulent Lewis number
$Le = 1$), the non-turbulent vertical flux is a methane bubble flux (see Section 2.7). The marginal
diffusive flux is calculated from condition of continuity of both concentration and flux at the water-
sediments interface (see more in Section 2.2), and $R_f$ represents only methane oxidation. Aero-
bic methane oxidation in water follows Michaelis-Menthen kinetics (14) with respective constants
$V_{max,w} = 1.16*10^{-5} \, \mathrm{mol}/(\mathrm{m}^3 * \mathrm{s})$ (Liikanen et al., 2002), $K_{CH_4,w} = 3.75*10^{-2} \, \mathrm{mol}/\mathrm{m}^3$ (Liika-
nen et al., 2002; Lofton et al., 2013), $K_{O_2,w} = 2.1*10^{-2} \, \mathrm{mol}/\mathrm{m}^3$ (Lidstrom and Somers, 1984).

### 2.6.3 Oxygen and carbon dioxide in water

Oxygen concentration is simulated by equation (4) with term $I$ neglected, assuming turbulent Lewis
number to be 1, and marginal diffusive flux treated as sedimentary oxygen demand (SOD). Other
sinks of oxygen are biochemical oxygen demand (BOD, excluding respiration), respiration and
methane oxidation. Methane oxidation bacteria consume oxygen according to widely-accepted stoi-
chiometric relation

$$CH_4 + 2O_2 = CO_2 + 2H_2O \tag{15}$$





providing the rates of oxygen consumption and $CO_2$ production given the rate of methane loss

(Section 2.6.2). The only process producing oxygen in a water column is photosynthesis. For biochemical oxygen demand, respiration and photosynthesis we use parameterizations from (Stefan and Fang, 1994). These parameterizations assume the rates of biogeochemical processes to depend exponentially on temperature and be proportional to chlorophyll-a concentration. Photosynthesis is additionally limited by photosynthetic active radiation. In our simulations, we kept the original em-

pirical constant values from (Stefan and Fang, 1994). For more details an interested reader may refer to the original paper.

As to sedimentary oxygen demand, we adopted the formulation from (Walker and Snodgrass, 1986), as it involves explicitly the near-bottom oxygen concentration (via diffusive term), in contrast to that from (Stefan and Fang, 1994), where SOD continues to be non-zero even when oxygen

content in water is nil.

Carbon dioxide in water is calculated by the same type of prognostic equation as for other gases. The only sink of $CO_2$ in the water column is photosynthesis, whereas its production in the model is provided by SOD, BOD, respiration and methane oxidation. As the rates of these processes in terms of $O_2$ and $CH_4$ are quantified above, the respective income or loss of $CO_2$ is immediately provided

by (15) and the following stoichiometric equality:

$$6CO_2 + 12H_2O + photons = C_6H_{12}O_6 + 6O_2 + 6H_2O, \quad - \text{ photosynthesis and respiration}, \qquad (16)$$

$$C + O_2 = CO_2, \quad - \text{ BOD and SOD}. \qquad (17)$$

### 2.6.4 Diffusive gas flux at the water-air interface

The top boundary condition (at the lake-atmosphere interface) for concentration of any dissolved gas

has the form:

$$k_s \frac{\partial C_w}{\partial z}\bigg|_{z=0} = F_{C_w}, \qquad (18)$$

where $C_w$ is $C_{CH_4,w}, C_{O_2,w}$ or $C_{CO_2,w}$, $k_s$ – dissolved gas diffusion coefficient and $F_{C_w}$ is the diffusive flux of a gas into the atmosphere, positive upwards. This flux is calculated according to the widely used parameterization:

$$F_{C_w} = k_{ge}(C_w|_{z=0} - C_{ae}), \qquad (19)$$





with $C_{ae}$ being the concentration of the gas in water equilibrated with the atmospheric concentration following Henry law and $k_{ge}$, m/s, denoting the gas exchange coefficient, the so-called "piston velocity". The latter is written as:

$$k_{ge} = k_{600}\sqrt{\frac{600}{Sc(T)}}, \tag{20}$$

with the Scmidt number $Sc(T)$ having individual values for different gases and being temperature-dependent. The $k_{600}$ coefficient has been a subject of numerous studies, and a number concepts have been proposed to quantify it (Donelan et al., 2002). We adopt two widespread options for $k_{600}$: (i) empirical dependence on wind speed and (ii) surface renewal model.

The dependency on wind velocity takes the form (Cole and Caraco, 1998):

$$k_{600} = C_{k_{600},1} + C_{k_{600},2}|\mathbf{u_{a,10}}|^{n_{k_{600}}}. \tag{21}$$

Here, $\mathbf{u_{a,10}}$ stands for the wind speed vector at 10 m above the water surface, $C_{k_{600},1} = 5.75 * 10^{-6}$ m/s, $C_{k_{600},2} = 5.97 * 10^{-6}$ $(m/s)^{1-n_{k_{600}}}$ are empirical constants. The simple empirical equation (21) "integrates" the effects of wind speed on a number of processes such as turbulence in adjacent layers of water and air, wave development and breaking, cool skin dynamics, and therefore

is likely to be not enough sophisticated to express adequately a wide variety of conditions met on real lakes. Therefore, we also included surface renewal model (MacIntyre et al., 2010; Heiskanen et al., 2014), that in terms of $k_{600}$ states:

$$k_{600} = \frac{C_{1,SR}(\epsilon|_{z=0}\nu_m)^{\frac{1}{4}}}{\sqrt{600}}, \tag{22}$$

where $\nu_m$ designates molecular viscosity of water, and $C_{1,SR} = 0.5$ is an empirical parameter. As

TKE dissipation rate is available directly from $k - \epsilon$ closure, we do not use any special parameterization for $\epsilon|_{z=0}$ as proposed in other works (e.g., (MacIntyre et al., 2010)).

### 2.7 Bubble model and its coupling to LAKE model

#### 2.7.1 Single bubble model

The bubble model used in LAKE closely follows that described in (McGinnis et al., 2006). Con-

375 sider the evolution of a bubble rising from the lake bottom, and consisting of a mixture of gases. The quantity of each $i$-th gas in the bubble, $M_i$, mol, changes due to its dissolution into the water according to equation:

$$\frac{dM_i}{dt} = v_b\frac{dM_i}{dZ} = -4\pi r_b^2 K_i(H_i(T)P_i - C_i), \ i = 1,...,n_g, \tag{23}$$





where $r_b$, m, is the bubble radius, $H_i$ - the Henry "constant" dependent on temperature $T$, K, $P_i$,
380    Pa, the partial pressure of $i$-th gas, $C_i$, mol/m$^3$, is the concentration of a gas in water, $K_i$ is the
exchange coefficient, $v_b$, m/s, is bubble vertical velocity, $Z$, m, is the vertical coordinate originating
at the bottom and pointing opposite to gravity, $n_g$ is the number of gases in a mixture.

Five gases are considered in a bubble: methane, carbon dioxide, oxygen, nitrogen and argon.
Water vapour constitutes minor contribution to a bubble pressure, and therefore neglected. Indeed,
the saturated vapour pressure at $20°$C is 23.4 hPA that is $\approx 2\%$ of atmospheric pressure. This is
the upper estimate for water vapour pressure contribution in bubbles, as the pressure increases with
depth, and saturation vapour pressure – decreases, due to water temperature decrease.

The temperature in the bubble is assumed to be equal to that of environmental limnetic water at
the depth of current bubble location, $Z$. It means that the heat exchange between the rising bubble
and water is expected to be intensive enough to dominate over the adiabatic cooling of the bubble. In
practical terms, this frees us from solving additional equation for bubble temperature. The tempera-
ture dependency of Henry constants for flat solution surface is taken from (Sander, 1999). The effect
of gas-water interface curvature on equilibrium gas pressure is omitted in this model because when
using Thomson (Kelvin) formula it turns out to be negligible for typical bubble radii in oceans and
lakes ($\geq 1$ mm). Exchange coefficient, $K_i$, is dependent on molecular diffusivity in water, bubble
radius and its velocity according to empirical formulae from (Zheng and Yapa, 2002). The bubble
velocity is determined assuming equilibrium between buoyancy force and environment resistance
given by quadratic law for small radii ($r_b < 1.3$ mm) and taking into account bubble surface oscil-
lations for larger sizes (Jamialahmadi et al., 1994).

For each component of gas mixture we apply an ideal gas law because under the typical pressures
at moderate water depths (at least dozens of meters) Van der Waals forces are small:

$$\frac{4}{3}P_i\pi r_b^3 = M_i RT, \; i = 1,...,n_g, \tag{24}$$

where $R$ is the universal gas constant. The surface tension pressure is small for the bubbles with
radii typical in lacustrine environment, and is neglected in (24). Then, when equating the gas mixture
pressure $\sum_{i=1}^{n_g} P_i$ to hydrostatic pressure at a given depth, $p_a + \rho_{w0}g(h_{bot} - Z)$ ($p_a$ the atmospheric
pressure, $h_{bot}$ is a lake depth in a point, where the bubble is released) and using (24) one yields:

$$r_b = \left[\frac{3RT\sum_{i=1}^{n_g} M_i}{4\pi(p_a + \rho_{w0}g(h_{bot} - Z))}\right]^{1/3}. \tag{25}$$





For solution of $2n_g + 1$ equations (23)-(25) the boundary conditions are needed. These are initial gases' molar quantities $M_{i,Z=0} = M_{i0}(t)$, $i = 1,...,n_g$, that are the quantities at the moment when the bubble crosses the lake bottom. In the model they are initialized as follows:

$$M_{i0} = \alpha_i M_0, \ i = 1,...,n_g,$$
$$M_0 = \frac{\frac{4}{3}\pi r_{b0}^3 (p_a + \rho_{w0} g h_{bot})}{R\, T|_{Z=0}}, \tag{26}$$

where $M_0$ - the total gas quantity in the bubble (mols). According to (26), the bubble initialization is provided by initial bubble radius, $r_{b0}$, and molar fractions of mixture components $\alpha_i$. In this study, we chose $r_{b0} = 2*10^{-3}$ m and the initial bubble gas composition to be 100% of $CH_4$.

The bubble model described above is numerically solved by Euler explicit scheme.

### 2.7.2 Bubble flux of gases

In equation (4) applied for methane, oxygen and carbon dioxide the non-turbulent flux (term $III$) consists of bubble flux only. Bubble flux also contributes to term $IV$ therein. This section explains how these terms are evaluated using single bubble model, described above (Section 2.7.1).

We consider an idealized situation when all bubbles rising from all columns of sediments have the same initial radius $r_{b0}$ at the bottom and identical gas composition. Given that in reality there is always a distribution of bubbles in their size, parameter $r_{b0}$ may be treated as an average (in appropriate sense) radius over this distribution. For bubbles rising from a given sediments column, equations (23)-(25) imply that their radius and composition will be the same at any level over this column.

Now, at any depth $z$ we can construct a horizontal average of vertical bubble flux of $i$-th gas, $\overline{F_{B,i}}(z) \approx A^{-1}(z) \sum_{k=1}^{n_s} F_{B,i,k}(z) A_k(z)$, where index $k$ is the index of a sediments column, $n_s$ being a total number of columns, and $A_k(z)$ is an area of projection onto $A(z)$ of the part of the top facet of $k$-th column residing below depth $z$ (e.g., for columns with tops above $z$, $A_k(z) = 0$; for columns of sediments with top facets completely below $z$, $A_k(z) = A_{s,k}$, $A_{s,k}$ standing for the area of top facet of $k$-th column). When the mean flux is calculated, it may be used in the term $III$ of (4)

$$+\frac{1}{A}\frac{\partial A\overline{F_{B,i}}}{\partial z}. \tag{27}$$

Here $F_{B,i}$ is defined as positive upwards leading to "+" sign.

To get the averaged flux $\overline{F_{B,i}}$ as described above, individual bubble fluxes $F_{B,i,k}$ are calculated from each sediments column as

$$F_{B,i,k} = M_{i,k} n_{b,k} v_{b,k}. \tag{28}$$



Here, we introduced bubble number density $n_{b,k}$, $\mathrm{m}^{-3}$, and $k$ is a sediments column index, as before. All bubbles that are released from a given sediments column's surface completely dissolve simultaneously at some depth or evade to the atmosphere. Furthermore, it is known that bubbles with diameter $\approx 1\ \mathrm{cm}$ are unstable and split up (Yamamoto et al., 2009; McGinnis et al., 2006). Hence, in the model it is assumed that a bubble with $r_b \geq 0.5\ \mathrm{cm}$ splits into two. In the depth interval between two subsequent bubble collapses the bubble flux (that is the number of bubbles crossing the horizontal surface of $1\ \mathrm{m}^2$ per second) is constant, and at the depth of division it doubles. Taking this into account, one may rewrite (28) as follows

$$F_{B,i,k} = F_{B,i,k}(h_{bot})N_k m_{i,k}, \tag{29}$$

where we have used the product $N_k m_{i,k}$ – bubble flux normalized by the bottom value, with $m_{i,k} = M_{i,k}/M_{i,k}(h_{bot})$, $N_k = (n_{b,k}v_{b,k})/(n_{b,k}(h_{bot})v_{b,k}(h_{bot}))$, and $F_{B,i,k}(h_{bot})$ standing for the the bubble flux at the bottom (top of $k$-th column of sediments). Evidently, $N_k(z) = 2^l$, $l$ being the number of bubble divisions happened below the depth $z$ over $k$-th sediments column. If the bottom bubble flux of one gas is known (in this model it is methane, $i = 1$, see Section 2.6.1) then the bottom fluxes of other gases are determined by bottom bubble composition

$$F_{B,i,k}(h_{bot}) = F_{B,1,k}(h_{bot})\frac{\alpha_i}{\alpha_1},\ i = 2,...,n_g. \tag{30}$$

### 2.8 Numerical aspects

The principal requirements for the numerical scheme of the diffusion-type model with nonlinear sources described above are integral conservation of prognostic variables and stability.

Integral conservation is achieved by employing second-order centered differences in space for all equations in water and sediments. The coupling of sediments columns to water body is also implemented ensuring continuity of heat and methane flux across the sediments-water interface.

Equations of $k - \epsilon$ closure are discretized in a way where TKE input by shear production and buoyancy in TKE equation equals to dissipation and potential energy source/sink in momentum and temperature/salinity equations, respectively (Burchard, 2002) (salinity is set to zero in this study).

Time-marching scheme is a Crank-Nicolson one (Crank and Nicolson, 1996) that allows for increased time steps, $\Delta t \approx 10\ \mathrm{min}$ for vertical grid spacing of $\approx 1\ \mathrm{m}$ in water, if not using surface seiche parameterization. The time step is limited due to high non-linearity of $k - \epsilon$ closure. However, the strongest constraint for time step arises when horizontal pressure gradients are calculated via mass conservation (7)-(10). These equations are solved by explicit scheme, and $\Delta t$ in this case should be less then the period of basin-scale surface seiche oscillations, estimated to be $\sim 1\ \mathrm{min}$ from Merian formula for Kuivajärvi lake.



Using Crank-Nicolson scheme in momentum equations allows for eliminating Coriolis terms in
kinetic energy equation.

The algorithmic implementation of the model numerical scheme is presented as a flowchart in Fig.
4.

## 3   The lake measurements and model setup

### 3.1   Measurements

Lake Kuivajärvi is a small (area 0.63 km²) boreal lake in Hyytiälä, Southern Finland (24°16' E,
61°50' N, 141 m above sea level) next to the well-established SMEAR II forest station (Station for
Measuring Ecosystem-Atmosphere Relations) (Hari and Kulmala, 2005). The lake has an elongated
shape extending about 2.6 km in North-West to South-East direction and having a maximal width of
400 m. The catchment area is 9.4 km² of mostly flat terrain with primary soil type of Haplic Podzol,
and the vegetation is mostly managed pine forest. The lake has a maximum and mean depth of 13.2
m and 6.4 m, respectively. Fluxes of momentum, sensible and latent heat are measured at 1.5 m
above the lake surface by eddy covariance (EC) technique. The measurement setup consisting of an
ultrasonic anemometer (USA-1, Metek GmbH, Germany) and an enclosed-path infrared gas analyzer
(LI-7200, LI-COR Inc., Nebraska, USA) is mounted on a fixed platform situated in the middle
of the lake. More details on the measurement platform, the EC system setup and flux calculation
procedures can be found in (Mammarella et al., 2015). On the platform, a four-way net radiometer
(CNR-1) provided the full radiation budget (shortwave and longwave) and a thermistor string of
16 Pt100 resistance thermometers (accuracy 0.2°C, depths 0.2, 0.5, 1.0, 1.5, 2.0, 2.5, 3.0, 3.5, 4.0,
4.5, 5.0, 6.0, 7.0, 8.0, 10.0 and 12.0 m) enabled the calculation of the heat storage in water and
the thermocline depth according to (Nordbo et al., 2011). All the atmospheric measurements were
performed at the height of 1.7 m above the water and 30-minute averages were calculated for the
analyses. In addition, the relative humidity (RH) was directly measured at the platform at the height
of 1.5 m (MP102H-530300, Rotronic AG, Switzerland). Manual water samplings for $CO_2$ and $CH_4$
were conducted weekly in the water column from the surface to the bottom (0.1, 1, 3, 5, 7, 9, 11 and
12 m). The $O_2$ content was measured every half-meter until the depths of 9 m and after that every
one meter (depths 0.1, 0.5, 1.0, 1.5, 2.0, 2.5, 3.0, 3.5, 4.0, 4.5, 5.0, 5.5, 6.0, 6.5, 7.0, 7.5, 8.0, 8.5, 9.0,
10.0, 11.0 and 12.0 m). These samples were processed using the headspace equilibrium technique as
described in (Miettinen et al., 2015). The used data are for the period 5-th of May to 31-st of October
2013.

### 3.2   Setup of numerical experiments

Numerical experiments with LAKE model were arranged in a way to fit the main objectives of
the study: (i) general assessment of model performance in temperature, $O_2$, $CO_2$ and $CH_4$, and



(ii) quantification of the role of lake stratification and turbulence regimes in vertical transport of gases. A set of experiments consists of a baseline (reference) model run and others, where physical

parameterizations or constants were varied.

The parameters of baseline experiment are given in Table 1. Maximal lake depth was set to 12.5 m to ease comparison with measurements, as this is the local depth below observational mast. There is no information on the lake sediments characteristics of Kuivajärvi Lake, however, silt loam should be close in soil particle size to typical lake sediments. Sediments depth (10 m) is chosen to be of

510 enough extent for temperature fluctuations not reaching its lower boundary. To get $A(z)$, we linearly interpolated the morphometric data given in Table 2.

Boundary conditions were set as follows. At the sediments bottom, zero heat and moisture fluxes were imposed. At the water surface, heat balance equation is applied, where downward radiation fluxes were measured at the mast, surface longwave radiation calculated via Stefan-Boltzmann law,

and sensible and latent heat fluxes – using Monin-Obukhov similarity functions (Table 1). In total, seven meteorological variables were supplied to the model, at 30 min interval, all measured at the lake: wind speed and direction, temperature, humidity, longwave and shortwave radiation, atmospheric pressure. For analysis of these time series, we refer to (Heiskanen et al., 2015).

Initial conditions for the model are the profiles of all prognostic variables at initial instant. Water

temperature, oxygen, carbon dioxide and methane vertical distributions were specified from measurements at 00:00 5 May 2013. Salinity was set to zero, two horizontal velocity components were initialized with small values. In sediments columns, temperature was set to $4°C$, and water content – to slightly undersaturated values.

Only two constants in the model were calibrated. The first one is $P_0$ in equation (12), controlling

the magnitude of methane production in sediments, representing quantity and "quality" of organics in sediments as a substrate for methanogenic activity. However, we found that this constant is not enough to regulate methane concentration in the lake mixed layer (see the rationale in Section 5.3). A half-saturation constant in $CH_4$ oxidation reaction rate, $K_{CH_4,w}$, was found to be crucial parameter in this respect, effectively changing mean levels of mixed-layer methane concentration.

The sensitivity experiments were set with the same configuration, as the baseline experiment, with the only modifications:

– surface seiches turned off (denoted hereafter as SS-)

– internal seiches parameterized via Goudsmit formulation (IS+)

– gravity waves parameterized with Mellor extension for $k - \epsilon$ model (GV+)

– internal seiches parameterized via Goudsmit formulation, surface seiches turned off (IS+SS-)

– gravity waves parameterized with Mellor extension for $k - \epsilon$ model, surface seiches turned off (GV+SS-)





Table 1: Parameters of baseline experiment

| | |
|---|---|
| Time span of integration | 5 May - 31 October 2013 |
| Time step, $\Delta t$ | 10 s |
| Vertical grid | 20 layers, refined near boundaries |
| Number of columns of sediments, $n_s$ | 5 |
| Vertical grid in columns of sediments | 10 layers, exponentially compacting towards sediments top |
| Physical parameters | |
| Albedo for visible radiation | 0.06 |
| Fraction of near infrared energy in shortwave flux | 35% |
| Water surface emissivity | 0.98 |
| Extinction coefficient for shortwave radiation | $0.58 \text{ m}^{-1}$ |
| Modal wind fetch | 410 m |
| Maximal lake depth, $h$ | 12.5 m |
| Vertical dimension of sediments columns, $h_{sed}$ | 10 m |
| Sediments (soil) type | Silt loam |
| Lake bottom effective roughness, $z_{0b,eff}$ | $10^{-3} \text{ m}$ |
| Initial bubble radius, $r_{b0}$ | $2 * 10^{-3} \text{ m}$ |
| Physical parameterizations | |
| Surface flux scheme | (Paulson, 1970; Businger et al., 1971; Beljaars and Holtslag, 1991) |
| Equation of state | (Hostetler and Bartlein, 1990) |
| Turbulence closure | standard $k - \epsilon$ with Canuto stability functions (Canuto et al., 2001) |

– minimal diffusivity in the thermocline increased (MD)

In the following sections we will describe and discuss main results of the baseline experiment and

sensitivity experiments in terms of physical and biogeochemical variables.





Table 2: Lake morphometry parameters

| Hypsometric curve | |
|---|---|
| Depth, $z$, m | Horizontal cross section area, $A(z)$, m$^2$ |
| 0 | $6.38 * 10^5$ |
| 1.5 | $5.41 * 10^5$ |
| 3 | $3.86 * 10^5$ |
| 6 | $2.27 * 10^5$ |
| 10 | $7.79 * 10^4$ |
| 12.5 | $7.0 * 10^3$ |
| Semi-major to semi-minor axis ratio of the elliptic lake shape, $L_x/L_y$ | 10 |

## 4 Results

### 4.1 Temperature and turbulent quantities

In this section we will consider temperature stratification and turbulent structure of the lake vertical column, that are prerequisites for correct simulation of biogeochemical processes. The surface mo-
mentum and energy fluxes will not be covered as they were discussed for this lake involving LAKE model results on these variables in (Heiskanen et al., 2015).

Evolution of temperature distribution in the lake is presented at Fig.5a (model) and Fig.5b (observations). Temperature profile at the beginning of May is nearly homogeneous at both figures, with values close to temperature of maximal density ($\approx 4^\circ$C). Then, as the net energy input in the lake
becomes positive, the surface mixed layer starts to heat up, achieving temperature values of above $22^\circ$C in both measurements and the model by mid-June. During summer, we may distinguish three periods of warm epilimnion ($> 22^\circ$C) interrupted by two cold periods ($< 18^\circ$C) that is caused by change of synoptic conditions in the atmosphere. Starting from the second part of August the net energy loss at the lake surface leads to mixed-layer cooling and eventually to homogenization of the
water column about $4^\circ$C.

The model satisfactorily reproduces the observed seasonal temperature pattern in Kuivajärvi Lake. The root mean square error (RMSE) for surface temperature is $1.54^\circ$C and the difference of means is $0.61^\circ$C (the average of modeled surface temperature slightly exceeds that of observed). However, a closer look into Fig. 5b reveals high-frequency fluctuations of observed temperature in the depth
range of the thermocline which are not reproduced by the model (Fig. 5a). These oscillations are of amplitude comparable with that of surface diurnal cycle at the surface, $1-2^\circ$C. We will address the nature and possible significance of these fluctuations in Section 5.1.



Fig.6 presents July-averaged profiles of TKE obtained in different model runs. In all model experiments the maximal amount of TKE is observed in the surface mixed layer, whereas the behaviour of
565 TKE below is different depending on the experimental setup. We see that in model runs with surface seiches switched off (marked by "SS-" in the legend) the minimal TKE is attained below thermocline, i.e. in the hypolimnion. In contrast to these, in model launches where barotropic pressure gradient was taken into account, TKE was produced below thermocline as well, while the TKE minimum is located inside the thermocline. Significantly, introducing Goudsmit internal-seiche-induced
mixing parameterization in the model (IS+ experiment) brings very small change to TKE profile. On the other side, Mellor gravity waves parameterization (Mellor, 1989) (GV+ experiment) adds considerable TKE to the profile of baseline experiment, especially in metalimnion and hypolimnion in conjunction with surface seiches taken into account.

### 4.2 Oxygen

Hereafter, for gases dissolved in water, we use concentrations per unit volume of water.

Dissolved oxygen evolution in Kuivajärvi Lake is presented at Fig.7a (model, reference experiment) and Fig.7b (observations). Here and in the subsequent plots for $CO_2$ and $CH_4$, we will confine our analysis to June-October period, as during May the modeled gas concentration undergo adjustment towards realistic patterns due to incorrect initial conditions in sediments. Large-scale features
in oxygen distribution agree in model and in measurements: maximal quantities of $O_2$ are concentrated in the mixed layer (since the photosynthesis rate is highest in the photic zone), whereas the minimal ones occur in the late summer near the bottom, due to consumption by sediments. For surface oxygen concentration, the averaged absolute bias is 1.27 mg/l (model value equals to 7.84 mg/l vs. the measured value of 9.11 mg/l), and the root mean square error (RMSE) is 1.37 mg/l.

Oxygen concentration is prominently different in the model from what was observed in the lake during spring and autumn turnovers. The model significantly underestimates oxygen levels below the mixed layer in spring and throughout the water column in autumn, by $\approx$ 3mg/l in the latter case. Another difference is in vertical gradient in hypolimnion: the model produces sharp gradient whereas in nature there is almost homogeneous oxygen distribution.

### 4.3 Carbon dioxide

Carbon dioxide concentration distribution in water is somewhat mirroring that of oxygen (Figs. 8a and 8b). The minimum of dissolved carbon dioxide is located in the mixed layer, while below the thermocline it is continuously accumulated during summer before reaching minimum throughout a water column at autumnal turnover. This general pattern is captured by the model.

Surface $CO_2$ density is considerably lower in the model compared to observations (time average being 0.39mg/l vs. 2.80 mg/l) with RMSE 2.35 mg/l. As in case of oxygen, the modeled $CO_2$ is characterized by high vertical gradients in hypolimnion, while the measured data demonstrates much



more homogeneous field. We also note overestimated hypolimnetic concentrations in the model until
September, when abrupt rise of $CO_2$ was detected by manual measurements.

### 4.4 Methane

Methane concentration in the lake water is low (Figs. 9a,9b), except for late September and beginning
of October when it increases near bottom up to 351.5 μg/l in the model and 536.0 μg/l according
to measurements. The model successfully reproduces this seasonal pattern, though it produces weak
maxima in $CH_4$ concentration close to sediments during summer. Observed maximum is single,
while in the model autumnal near-bottom methane rise is disrupted by sharp decrease in beginning
of October, leading to two concentration peaks. The surface concentration remains small through
all the simulation time, with mean value 0.89 μg/l in the model and 1.06 μg/l in observations and
model RMSE being 0.83 μg/l.

Due to the low methane amount in Kuivajärvi Lake surface waters the flux of this gas to the atmo-
sphere is negligible. Average eddy covariance $CH_4$ flux (not shown) is only 0.0006 μmol/(m$^2$s) (0.8
mg/(m$^2$day)). The average diffusive flux at the lake surface in model is 0.0005 μmol/(m$^2$s) (0.7
mg/(m$^2$day)). Whereas diffusive flux in the LAKE model can be treated as an average one over the
water body surface, bubble flux at this surface is calculated over each sediments column separately
(Sect.2.7), i.e. it is different over different lake depth zones. Therefore, to compare the total methane
flux (diffusive plus ebullition) to eddy covariance measurements, it is the bubble flux from the sedi-
ments column locating approximately below the EC footprint that should be used (for EC footprint
at Kuivajärvi Lake, see (Mammarella et al., 2015)). In our case, it is the deepest sediments column,
where the time average $CH_4$ ebullition flux reaching the surface constituted 0.006 μmol/(m$^2$s) (8
mg/(m$^2$day)). Thus, the mean total methane flux in the model exceeded the observed one by an
order of magnitude, still remaining low compared to that at many other lakes (Juutinen et al., 2009).

## 5 Discussion

### 5.1 Temperature

#### 5.1.1 Overview of the model performance in temperature

The three-layer stratification in the lake (epilimnion, metalimnion, hypolimnion) is well reproduced
by the LAKE model. This is the most significant summertime feature impacting the distribution of
all physical quantities in a lake as well as of biogeochemical ones. In this situation, the applicability
of 1D approach is facilitated by extremely stable stratification in the thermocline (in Kuivajärvi Lake,
metalimnetic Brunt-Väisälä frequency exceeded 0.1 s$^{-1}$ in mid-summer), that is the typical feature
of dimictic lakes compared to large and deep lakes and oceans. This stratification is a key dynamic
factor to which other ones have to be compared. Specifically, the wind force impact, disturbing the



lake's layered structure, is assessed via Wedderburn number ($W$) (Shintani et al., 2010) or Lake number (Imberger and Patterson, 1989), while the significance of Coriolis force can be quantified by comparing Rossby deformation radius ($L_R$) to a lake size (Patterson et al., 1984). The two parameters, $W$ and $L_R$, are plotted in Figs. 10 and 11, respectively. We see that during June-August

$W$ generally fluctuates around 50 implying that thermocline vertical displacement by wind forcing is about $\sim 100$ times less than the mixed-layer depth. However, in May, end of October and on short periods of several days during summer $W$ approaches unity making possible upwelling at lake's margins, similar to what was reported for 2011 in (Heiskanen et al., 2014). At least two of these episodes, namely, these in mid-June and end of July, are concomitant to mixed-layer cooling (Fig.

5a), weakening the lake stratification.

Rossby deformation radius, $L_R$, is similar or smaller than the lake length ($\approx 2600$ m), so that Coriolis force should significantly modify the currents here. Indeed, neglecting Coriolis force from dynamic equations of the model drastically increased vertical mixing in our simulations (not shown), making the mixed-layer depth unrealistically large.

The surface temperature time series are realistically reproduced by the model. This result is achieved by both the high quality of atmospheric forcing (all atmospheric variables were measured over the lake surface) and the properties of the model used. Surface temperature is defined by net heat stored in the mixed layer and the mixed-layer depth, i.e. the depth over which this heat is distributed. Hence, the parameterization of sensible and latent heat fluxes (the only unknowns in the net

heat when radiation fluxes are measured), and momentum flux (the primary source for TKE production in the mixed layer) are critical for calculating surface temperature correctly. Serious concerns on validity of Monin-Obukhov similarity theory for a case of lake surrounded by bluff topography have been reported in literature (e.g., see reasoning based on results of LES (Glazunov and Stepanenko, 2015) or laboratory experiments (Markfort et al., 2013)). However, it turns out in practice of 1D

lake model applications to such lakes, that not only Monin-Obukhov is the most physically-based option in these models for obtaining surface heat fluxes so far, but it is that still delivering acceptable accuracy for calculated surface temperature at seasonal timescales (Stepanenko et al., 2014; Heiskanen et al., 2015). As to momentum flux, it has been shown to be a crucial parameter regulating the rate of mixed-layer deepening during summer (see e.g. (Stepanenko et al., 2014)). This resulted in a

widespread modelling practice where the drag coefficient, defining momentum flux at a given wind-speed, has become a tunable parameter in $k - \epsilon$–based lake models. In our simulations, we do not calibrate surface drag coefficient, but include a simple parameterization of momentum flux partitioning between waves and currents (Stepanenko et al., 2014), leading to reduction of mixed-layer depth towards observed values (not shown).





### 665  5.1.2  Internal seiches

Consider now in more detail temperature fluctuations in the thermocline, not reproduced by the model (cf. Figs. 5a and 5b). Measured time series of temperature demonstrate, that these fluctuations appear below the base of mixed layer, where diurnal temperature variability diminishes. Thus, they are caused by neither diurnal cycle of surface net heat nor by shortwave radiation absorption in the
water column. Their occurrence throughout a thick (about 4 m) layer of stable stratification, with Richardson number, $\mathrm{Ri} \gg 1$ (according to model results), where the vertical eddy conductivity should be largely suppressed, hints at the only feasible mechanism of these temperature changes that is due to (organized) vertical advection. Thence, the periodic character of these fluctuations implies flow oscillations, i.e. internal waves.

Fig.12 shows Fourier spectrum of temperature time series at three depths in the thermocline. At all depths there are two distinct maxima at frequencies: $\omega \approx 8.5*10^{-5}$ s$^{-1}$ ($T_{seiche} \approx 20.5$ hour) and $\omega \approx 4.5*10^{-4}$ s$^{-1}$ ($T_{seiche} \approx 3.9$ hour). The harmonic of $T_{seiche} \approx 20.5$ hour contains much more energy than that of $T_{seiche} \approx 3.9$ hour. In order to interpret these spectra we use the method for seiche period calculation proposed by (Münnich et al., 1992).

Starting from two-dimensional linearized incompressible Boussinesq equations and seeking the solution for vertical velocity, $w$, in a wave-like form:

$$w(x,z,t) = W(z)\exp[i(kx - \omega t)] \tag{31}$$

with rigid lid condition $w|_{z=0,h} = 0$ leads to an ordinary differential equation for the amplitude, $W$:

$$\frac{d^2 W}{dz^2} + \left(\frac{N^2}{\omega^2} - 1\right)k^2 W = 0, \tag{32}$$

$$W|_{z=0,h} = 0, \tag{33}$$

which is a Sturm-Liouville problem for frequencies, $\omega$ and corresponding eigenfunctions, $W$. We solved it by shooting method with squared Brunt-Väisälä frequency $N^2$ taken from the mean temperature profile measured in July and $h = 12.5$ m (a depth of lake in the point of measurements). Considering 1-st horizontal mode, $k = \pi/L_{x0}$, we've got $T_{seiche,1} = 6.5$ hour with $W$ having a
form of the 1-st vertical mode, usually denoted as V1H1 (one maximum of $W$ between $z = 0$ and $z = h$) and $T_{seiche,2} = 21.2$ hour for the 2-d vertical mode (one maximum of $W$ and one minimum, V2H1). These frequencies correspond to those of maxima at the temperature spectrum (Fig.12). The discrepancy between measured and calculated frequencies, that especially noticeable for V1H1 mode (3.9 hour vs. 6.5 hour, respectively), is expectable since the linear analysis described above
neglects morphometry of the lake's bed (Fricker and Nepf, 2000), effects of Coriolis force and the complex temporal behaviour of the actual wind forcing.





The prominence of V2H1 mode in the temperature spectrum is what have been found for an Alpine lake by M.Münnich as well (Münnich et al., 1992). A plausible explanation for that is the resonance between V2H1 seiche and the wind speed, both having close to diurnal periodicity (Mortimer, 1953).

Thus, the main conclusion of this section is a presence of significant internal seiches in Kuivajärvi Lake that may be responsible for additional mixing in the thermocline either in the interior of the lake or at its margins. This will be discussed in the following section (Section 5.2).

### 5.2  Turbulent quantities

In this section we will focus on turbulence characteristics in the thermocline and hypolimnion as 705 they are factors for vertical transport of gases originating at a lake bottom. Moreover, the presence of seiches in the lake suggests additional mixing mechanisms to exist in the thermocline, such as production of TKE by near-bottom shear (Goudsmit, 2002) and breaking of internal waves at the sloping bed (MacIntyre et al., 2009; Boegman et al., 2005).

### 5.2.1  TKE production terms

The vertical distribution of TKE shown at Fig.6 is formed as a result of approximate balance between terms in right-hand side of TKE equation (B1). Mean vertical distribution of TKE production by shear, $S$, by buoyancy, $B$, and by seiches, $S_{seiche}$ (only when Goudsmit parameterization is used, "IS+" experiment) in July is shown in Fig.13.

First, we see that mean buoyancy production is positive in the top half of mixed layer ($\sim 10^{-9} \div$ 715 $10^{-7}$ m$^2$s$^{-3}$), indicating that nocturnal buoyancy production of TKE in this region overrides the daytime sink. It is several times (up to an order of magnitude) less than the shear production, however, exceeds 3-5 orders of magnitude generation of TKE by seiches. Different experiments show almost identical profiles of $B$. It is because both Goudsmit and Mellor parameterizations include dependence on $N$ providing zero contribution to TKE and other turbulent quantities at $N = 0$, and 720 $N \approx 0$ in the mixed layer. Below, buoyancy production becomes negative due to stable stratification.

Vertical shear production is the largest contributor to TKE throughout a lake profile excepting thermocline, at $\sim 7$ m depth, where it attains its minimum and becomes less than $S_{seiche}$. This minimum corresponds to TKE minimum (Fig.6) and a minimum of eddy viscosity, $\nu$, approaching minimal value, $\nu_{min}$. As in the model we do not use any "background diffusivity/viscosity/conductivity", the 725 minimum value of $\nu$ and $\nu_T$ is set to a very small number, $\nu_{min} = \nu_{T,min} = 10^{-8}$ m$^2$/s (cf. molecular viscosity at $10°$C, $\nu_m = 1.307 * 10^{-6}$ m$^2$/s and heat diffusivity, $\nu_{T,m} = 1.41 * 10^{-7}$ m$^2$/s). Hence, $S = \nu[(\partial u/\partial z)^2 + (\partial v/\partial z)^2]$ reaches negligible values, as $\nu = \nu_{min}$. Below thermocline, there is drastic difference in $S$ between experiments where dynamic barotropic pressure gradient was taken into account (baseline experiment), and those without surface seiches – labelled by "SS-" 730 on Fig.13. The reason is that due to water surface inclination, currents are generated in hypolimnion, while stratification is not strong enough to dominate over shear (Ri $< 0.25$, not shown). The largest



shear production takes place in the experiment with both Mellor parameterization and dynamic pressure gradient included ("GV+"). The value of $S$ is especially increased in the thermocline, because $N^2$ reaches maximum there, and it contributes to corresponding additional shear proportionally.

TKE also achieves maximal values for this experiment at all depths (Fig.6). Still, heat conductance and diffusivity in the metalimnion are close to molecular values even in this case.

Additional shear production due to seiches attains maximum in the thermocline with minima in the epilimnion and hypolimnion. This is again due to proportionality $S_{seiche} \propto N^2$. The contribution of $S_{seiche}$ to TKE production remains minor compared to shear everywhere, excepting a small region in

the thermocline, where TKE generation by vertical shear plunges to minimum, as discussed above.

The strong effect of surface seiches on under-thermocline turbulence obtained in our study is yet to be verified with more complicated models (i.e. 3D Reynolds-Averaged Navier-Stokes or Large Eddy Simulation) and extensive turbulence measurements for Kuivajärvi Lake. Indeed, surface seiches are barotropic motions not taking into account density stratification in a lake. As a consequence, their

period of $\sim 1$ min for Kuivajärvi Lake is orders of magnitude less than that of V1H1 mode (6.5 h) and higher modes obtained from eigenvalue problem for continuous stratification (Section 5.1). Taking into account internal seiches in the model would change drastically frequencies of near-bottom current oscillations compared to surface seiches and thereby the hypolimnetic shear production of TKE. However, so far, to the best of our knowledge, internal seiche parameterization producing

extra mixing in the hypolimnion has not been developed, as the poineering attempt by (Goudsmit, 2002) introduced $S_{seiche} \propto N^2$, negligible in hypolimnion. Envisaging implementation of internal motions in the model for our future work we, however, note that introducing surface seiches allowed to generate TKE below thermocline qualitatively consistent with a bulk of observational data (Wüest et al., 2000; Wüest and Lorke, 2003), demonstrating that summer stratification in dimictic lakes is

comprised of two turbulent layers disconnected by quasi-laminar thermocline.

### 5.2.2 Stationary Richardson number

Stationary Richardson number, $\mathrm{Ri}_{st}$ have been used in a number of studies ((Burchard, 2002) and references therein), to characterize maximum stability under which $k - \epsilon$ still model does not decrease TKE. Formally, it is a value of Ri derived from $k - \epsilon$ model under homogeneous and stationary

conditions. For standard $k - \epsilon$ model, it takes the form:

$$\mathrm{Ri}_{st} = \frac{N^2}{M^2} = \mathrm{Pr}\frac{\Delta c_{\epsilon 21}}{\Delta c_{\epsilon 23}}, \tag{34}$$

where $M^2 = [(\partial u/\partial z)^2 + (\partial v/\partial z)^2]$ is a shear frequency squared, Pr is turbulent Prandtl number, $\Delta c_{\epsilon 21} = c_{\epsilon 2} - c_{\epsilon 1}$, and $\Delta c_{\epsilon 23} = c_{\epsilon 2} - c_{\epsilon 3}$. With constants, $c_{\epsilon 1} = 1.44$, $c_{\epsilon 2} = 1.92$, $c_{\epsilon 3}$ is switched between two values depending on stratification, $c_{\epsilon 3} = 0.5 * [1 - \mathrm{H}(B)] * (-0.4) + 0.5 * [1 + \mathrm{H}(B)] *$

$1.14$, $\mathrm{H}()$ − Heaviside function, ensuring $c_{\epsilon 3} = -0.4$ in stable stratification and $\mathrm{Ri}_{st} = 0.25$.





Introducing additional shear by gravity waves into the total shear (Mellor, 1989), $S = \nu(M^2 + \alpha_g N^2)$, in both TKE and $\epsilon$ equations, doing analogous algebra as for (34), leads to modification of stationary Richardson number:

$$\mathrm{Ri}_{st} = \mathrm{Pr}\frac{\Delta c_{\epsilon 21}}{\Delta c_{\epsilon 23} - \alpha_g \mathrm{Pr}\Delta c_{\epsilon 21}}, \qquad (35)$$

yielding, with $\alpha_g \approx 0.7$ (Mellor, 1989), an increased estimate, $\mathrm{Ri}_{st} = 0.32$.

On the other side, when $k-\epsilon$ model is supplemented by Goudsmit internal seiche parameterization (Goudsmit, 2002), i.e. when the shear production is modified as $S^* = S + S_{seiche}$, $S = \nu M^2$, an expression for stationary Richardson number may be derived as well (see Appendix D):

$$\mathrm{Ri}_{st} = \frac{\mathrm{Pr}\Delta c_{\epsilon 21}}{\Delta c_{\epsilon 23} - \nu_0^{-1}\mathrm{Pr}C_s\Delta c_{\epsilon 21}(u_a^2 + v_a^2)^{3/2}}, \qquad (36)$$

$(u_a, v_a)$ standing for wind vector in the surface layer, $\nu_0$ - eddy viscosity at stationary turbulence regime, $C_s$ - constant for a given lake including empirical parameters and lake morphometry characteristics. As there are no unique values of $k$, $\epsilon$ and $\nu_0$ resulting from uniformity and stationarity conditions, we assume a small value $\nu_0 \approx \nu_m$ leading to an upper estimate, $\mathrm{Ri}_{st} = 0.30$. Larger values of $\nu_0$, according to (36), would decrease $\mathrm{Ri}_{st}$.

Estimates provided above suggest that $\mathrm{Ri}_{st}$ in $k-\epsilon$ model still remains under unity, when gravity waves and internal seiche parameterizations are included. Thus, they cannot generate significant turbulence in the thermocline of Kuivajärvi Lake, where $\mathrm{Ri} >> 1$. Indeed, in all experiments minimal eddy diffusivity in the thermocline was close to minimal possible one set in the code, $10^{-8}$ m$^2$/s, implying only molecular diffusion to perform vertical transport. Still, we envisage a possibility of mixing mechanisms rising total diffusivity above molecular levels in the metalimnion, given empirical evidences (e.g. (Saggio and Imberger, 2001)) and the fact that Mellor and Goudsmit parameterizations have not been tested thoroughly vs. extensive measurement data and/or LES and DNS simulation so far. Therefore, we conducted a sensitivity test on the influence of artificially increased diffusivity in the thermocline on gas concentrations ("MD" experiment, see next Section, 5.3).

### 5.3 Oxygen, methane and carbon dioxide

As we see at Figs 7a and 7b oxygen concentration is high in beginning of June not only in the mixed layer ($8-9$ mg/l), where it is produced by photosynthesis, but beneath the mixed layer as well ($5-7$ mg/l). This is due to maximal oxygen concentrations throughout a water column during the spring overturn in beginning of May. Afterwards, oxygen remains high in the mixed layer while it decreases to almost zero values in hypolimnion by August.

A conspicuous feature of $O_2$ content modeled is its gradual decline in the mixed layer during the deepening of the latter throughout October, from $\approx 7$ mg/l to $\approx 5$ mg/l, whereas observed values



even increased up to $\approx 9$ mg/l. In the model, oxygen production due to photosynthesis reduced by beginning of autumn under drop of photosynthetically active radiation, and mixed-layer deepening

caused dilution of oxygen amount over a larger volume, reducing concentration. As to a rationale for the oxygen concentration rise in measurement data, we postpone it for the future research. However, we can expect the change of phytoplankton communities when passing from summer stratification to autumnal mixing, and that these communities have different parameters of photosynthesis-irradiance (P-I) curve. These effects has not been included in the model so far.

The process of oxygen depletion in hypolimnion occurs differently in nature and in the model: in the model the rate of oxygen depletion increases with depth, causing significant vertical concentration gradients, while in the measured field there is almost homogeneous distribution over depth, i.e. the rate of oxygen decrease is near constant with depth. This discrepancy may be due to misrepresentation in the model of two processes: vertical diffusion and biogeochemical oxygen consumption

(sedimentary oxygen demand and biochemical oxygen demand).

In the model, BOD is distributed with depth according to temperature dependence only, so that it decreases towards the deepest point. In contrast, SOD, due to originating at lake margins, is represented as a marginal flux (see Section 2.1), i.e. being $\propto A^{-1} dA/dz$. Hence, SOD rises to maximum value as $A \to A_h$, $A_h = A(z = h)$. In hypolimnion, BOD $\sim 10^{-9}$ mol/(m$^3$s) during summer

815   months, while SOD $\sim 10^{-8}$ mol/(m$^3$s) increasing from $\approx 1 * 10^{-8}$ mol/(m$^3$s) at the top of hypolimnion to $\approx 6 * 10^{-8}$ mol/(m$^3$s) at its base. It is reasonable to expect the same morphometrical effect on SOD in nature, but it should superimpose at SOD dependence on temperature and biogeochemical characteristics of sediments, that are depth dependent as well.

Unfortunately, so far, there is no observational data for Lake Kuivajärvi (e.g. turbulence mea-

surements or any sediments data), facilitating to discern, whether it is enhanced turbulence below thermocline and/or nearly homogeneous SOD distribution with depth that makes measured oxygen profiles to be much more even than these in the model.

Consistently, the same questions arise considering carbon dioxide distribution (Figs 8a and 8b). Neglecting the spot of low $CO_2$ hypolimnetic amount in the measured pattern around mid-August,

that might be due to measurement errors, we see larger uniformity in the measured vertical distribution than in that calculated. Bottom concentration rises much faster in the model up to $\approx 16$ mg/l by mid-August, whereas in observed field this level is attained by mid-September only. This fast bottom accumulation of calculated $CO_2$ corresponds to fast decrease of $O_2$ (Fig. 7a). This corroborates our suggestion above that either vertically even SOD or vertical mixing (or both) are misrepresented

in the model under thermocline, as these processes affect $CO_2$ and $O_2$ in the way to homogenize hypolimnetic profile.

We also note that abrupt increase of deep $CO_2$ concentration that took place according to measurements in September in the depth interval 8–12 m is absent in the model. We argue that this rise is unlikely to be caused by local aerobic decomposition of organic matter, as the oxygen is depleted



near bottom by this time, $< 1\ \mathrm{mg/l}$ (Fig. 7b), and this amount is far from enough to contribute
to $CO_2$ jump by $5 - 7\ \mathrm{mg/l}$, given stoichiometric ratio or corresponding reactions $O_2 : CO_2 \sim 1$.
Hence, we suppose this $CO_2$ is advected to the point of measurements from catchment. Moreover,
this early autumnal sharp increase of carbon dioxide is likely to be a peculiarity of 2013; at least,
in 2011 and 2012 rising $CO_2$ hypolimnetic concentration was much more smooth (Miettinen et al.,
2015).

Kuivajärvi Lake is a significant source of $CO_2$ (Heiskanen et al., 2014; Miettinen et al., 2015;
Mammarella et al., 2015), and significant underestimation of surface concentration by the model
($0.39\ \mathrm{mg/l}$ vs. $2.80\ \mathrm{mg/l}$ measured) is a serious drawback of the model setup. As carbon dioxide
in the mixed layer is affected by a large number of processes (BOD, SOD, respiration, photosyn-
thesis, diffusion to the atmosphere), it seems for us difficult to disentangle this problem on a solid
physical/biogeochemical basis in this study, and it should be a part of separate research.

As stated above, only two constants were calibrated in the model, i.e. $P_0$ and $K_{CH_4,w}$, that are
responsible for magnitude of methane production in sediments and methane oxidation in water,
respectively. The value $P_0 = 3 * 10^{-8}\ \mathrm{mol/(m^3 * s)}$ chosen occurred to be very close to the value
obtained for thermokarst Shuchi Lake in North-Western Siberian ($P_0 = 2.55 * 10^{-8}\ \mathrm{mol/(m^3 * s)}$,
see (Stepanenko et al., 2011)). We note that it is not straightforward to compare these values, because
the model version used in Shuchi Lake study, lacked such important features as taking into account
bottom morphometry, bubbles dissolution in water, all biogeochemical processes involving $O_2$ and
$CO_2$ but methane oxidation. Nevertheless, the same order of magnitude of $P_0$ for two lakes of
different genetic types with ecosystems functioning under drastically different climate conditions,
argues for robustness of our model formulation. A half-saturation constant for methane, $K_{CH_4,w} =$
$3.75 * 10^{-2}\ \mathrm{mol/m^3}$ was set close to upper estimate of this parameter, found in literature (Martinez-
Cruz et al., 2015).

Due to high oxygen content, Kuivajärvi Lake is generally poor in methane (Miettinen et al., 2015).
To better understand the reasons for low surface methane concentrations, it is instructive to scrutinize
the budget of methane in the mixed layer (Fig. 14). Wee see that the methane fluxes nearly compen-
sate each other, bubble fluxes being dominant in magnitude (see Table 3). Divergence of bubble
flux is almost compensated by oxidation, whereas diffusion through thermocline is the smallest flux.
Thus, in the model, the epilimnetic and hypolimnetic pools of methane are almost "disconnected"
due to minimal TKE in metalimnion (see Section 5.2). Moreover, the total methane influx from shal-
low sediments is $\approx 6$ times larger than methane input by bubbles from deep sediments, i.e. those
below mixed layer ($23.22$ vs. $4.63\ \mathrm{mg/(m^2 day)}$). This implies that shallow sediments are the main
contributor of methane to the mixed layer, so that surface $CH_4$ concentration and eventually its
diffusive flux to the atmosphere are controlled by methane production in shallow sediments and
epilimnetic oxygen amount (via oxidation). However, bubble methane flux from deep sediments is





a considerable part of the total $CH_4$ flux to the atmosphere, since in the model 68-70% of methane leaving sediments at depth 12.5 m in bubbles, reaches the surface.

In the numerical experiment "SS-" with LAKE model, where surface seiches (horizontal pressure gradient) were neglected, the seasonal pattern of methane concentration took the form presented at Fig. 9c. In this case, the basal methane content began to rise about 2 months earlier than it was observed (Fig. 9b) and calculated in the reference run (Fig. 9a) and reached maximal value of 598.5 mg/l vs. 351.5 mg/l in baseline experiment. This is caused by earlier $O_2$ depletion (not shown) due to negligible oxygen supply from above waters in conditions of very small TKE in hypolimnion (Fig. 6). Hence, we conclude that hypolimnetic turbulence is significant for gases accumulation and vertical distribution there, although it is likely to be of minor importance for mixed-layer concentrations of $O_2$, $CO_2$ and $CH_4$, because of small gas transfer through metalimnion (see Section 5.2).

Table 3: Mean for methane fluxes in/out the lake mixed layer, $mg/(m^2 * day)$, normalized by lake surface area, May – October 2013. Positive terms are these transporting $CH_4$ into the mixed layer

| | |
|---|---|
| Diffusion at the lake surface | -0.86 |
| Diffusion at the bottom of mixed layer | 0.09 |
| Diffusion plus ebullition from mixed-layer sediments | 23.22 |
| Ebullition at the bottom of mixed layer | 4.63 |
| Ebullition at the lake surface | -20.31 |
| Oxidation in the mixed layer | -7.41 |
| Residual (storage change) | -0.64 |

Finally, as the complete dissipation of turbulence under strong stratification is questioned by a number of lacustrine observational studies (Saggio and Imberger, 2001) and theoretical considerations (Zilitinkevich et al., 2012), we conducted a model run "MD" with increased minimal eddy viscosity, diffusivity and heat conductance, i.e. $\nu_{min} = \nu_{T,min} = 10^{-6} \ m^2/s \approx \nu_m \approx 10\nu_{T,m}$. This 10 times molecular diffusion through thermocline lead to drastic decrease in methane concentration below, so that the maximal bottom amount from June to October attained only 48.38 μg/l vs. 351.51 μg/l in a reference experiment. It was caused by enhanced downward diffusion of oxygen from the mixed layer, consequently oxidizing methane diffused from sediments. Therefore, even suppressed turbulence may cause significant impact on hypolimnetic concentration of gases, having implications not only for greenhouse gases but also for anoxia events.

## 6 Conclusions

In this study a new version of 1D lake model LAKE is presented. It solves equations for temperature, momentum, turbulent kinetic energy and its dissipation rate, oxygen, carbon dioxide and methane





in a generic form derived for horizontally averaged arbitrary prognostic variable. Heat and methane
vertical transport are additionally realized in a set of vertical sediments columns that are coupled
to a water body via continuity of flux and temperature (concentration). The fluxes of momentum,
oxygen and carbon dioxide at the sloping bottom are described by appropriate formulations basing
on boundary layer laws and in-sediments biogeochemistry. The key biogeochemical transformations

between $O_2$, $CO_2$ and $CH_4$ in water are implemented. Both diffusive and ebullition flux of all gases
are take into account. Standard $k - \epsilon$ turbulence closure is supplemented by parameterizations of in-
ternal seiches (Goudsmit, 2002), gravity waves (Mellor, 1989) and a new surface seiche formulation,
following original concept by U.Svensson (Svensson, 1978).

     The model is validated vs. extensive measurement data collected by University of Helsinki at

Kuivajärvi Lake (Southern Finland) (Miettinen et al., 2015; Mammarella et al., 2015) during ice-
free season of 2013 and including all meteorological variables above lake surface necessary to drive
the model. In-water temperature, $O_2$, $CO_2$ and $CH_4$ vertical profiles from the water column served
to validate the model output.

     The model was successful in capturing large-scale patterns of spatio-temporal variability of tem-

perature and gases. In all the model parameterizations, only two constants relevant to $CH_4$ produc-
tion and consumption were calibrated. The value of $P_0$, regulating methane production in sediments,
occurred to be very close to those obtained in our previous study for a thermokarst lake in North-
Eastern Siberia (Stepanenko et al., 2011), corroborating the robustness of the model used. It is un-
certainty in a number of other parameters, responsible for reactions involving $O_2$ and $CO_2$, that is

likely to contribute to model errors in hypolimnion and these of $CO_2$ in the surface layer.

     As both carbon dioxide and methane typically accumulate below metalimnion in freshwater lakes
(e.g. (Bastviken et al., 2008)), the vertical transport of these gases below mixed layer becomes
an important factor for their evasion to the atmosphere. Our experiments together with stationary
Richardson number analysis show that Mellor and Goudsmit extensions of $k - \epsilon$ model neither pro-

duce TKE in hypolimnion, nor generate turbulence in thermocline enough to sustain eddy diffusivity
above molecular constant. However, surface seiche parameterization allowed to produce turbulence-
enhanced hypolimnion qualitatively consistent with empirical knowledge so far (Wüest et al., 2000;
Wüest and Lorke, 2003). Reproducing considerable TKE in hypolimnion lead to much better corre-
spondence of calculated $CH_4$ to observed one.

As there are strong doubts on complete suppression of turbulence even at Ri $\gg 1$ (Saggio and
Imberger, 2001; Zilitinkevich et al., 2012), we conducted an experiment with increased minimal
diffusivity in thermocline, 10 times the molecular coefficient, causing multifold decrease in near-
bottom methane concentration. This points at thermocline turbulence to be a crucial bottleneck in
quantifying greenhouse gas budget in lakes.

To conclude, we emphasize a role of internal lake oscillations and possible thermocline turbulence
in vertical transfer of dissolved gases. These factors are omitted in majority of lake models developed





so far, and should be addressed carefully in their future formulations. This will allow to get more rigorous regional and global estimates of greenhouse gases evasion to the atmosphere.

## 7   Code availability

The code of LAKE 2.0 model is available on request from the author (Victor Stepanenko, stepanen@srcc.msu.ru, vstepanenkomeister@gmail.com). The code is supplied by Makefile to ease the compilation under Linux, the technical documentation and users manual are provided in the model archive as well.

*Acknowledgements.*   This research is implemented in framework of Russian-Finnish collaboration, funded within
CarLac (Academy of Finland, 1281196) and GHG-Lake projects. Russian co-authors are partially supported by grants of Russian Foundation of Basic Research (RFBR 14-05-91752, 15-35-20958). In addition, Academy of Finland Centre of Excellence (118780) and Academy Professor projects (1284701 and 1282842); ICOS (271878), ICOS-Finland (281255) and ICOS-ERIC (281250) and the Nordic Centre of Excellence – DEFROST are acknowledged.



## 8 List of symbols

### 8.1 Thermodynamics and hydrodynamics

| | |
|---|---|
| $\lambda_m = \nu_{T,m} c_w \rho_{w0}$, J/(m $*$ s $*$ K) | molecular heat transfer (conductance) coefficient |
| $\lambda_t = \nu_T c_w \rho_{w0}$, J/(m $*$ s $*$ K) | turbulent heat transfer (conductance) coefficient |
| $\Delta t$, s | model time step |
| $\Gamma_{A(z)}$ | the boundary of a horizontal cross-section of a lake at depth $z$ |
| $\xi = z/h$, n/d | normalized vertical coordinate, pointed along gravity |
| $\rho_{w0} = 1000$ kg/m$^3$ | reference water density |
| $\epsilon$, m$^2$/s$^3$ | TKE dissipation rate |
| $\omega$, s$^{-1}$ | frequency |
| $A(z)$, m$^3$ | the area of horizontal cross-section of a lake at depth $z$ |
| $\nu$, m$^2$/s | turbulent viscosity in water |
| $\nu_m = 1.307 * 10^{-6}$ m$^2$/s | molecular viscosity of water |
| $\nu_T$, m$^2$/s | turbulent temperature transfer coefficient in water |
| $\nu_{T,m} = 1.41 * 10^{-7}$ m$^2$/s | molecular temperature transfer coefficient in water |
| $B$, m$^2$/s$^3$ | buoyancy production/sink of TKE |
| $B_s$, m/s | precipitation minus evaporation at a lake surface |
| $c_w = 3990$ J/(kg $*$ K) | water specific heat |
| $\mathbf{F} = \{F_1, F_2, F_3\} = \{F_x, F_y, F_z\}$ | non-advective (turbulent and non-turbulent flux) a state variable $f$ |
| $F_{nz}$ | non-turbulent vertical flux of a property $f$ |
| $F_{tz}$ | turbulent vertical flux of a property $f$ |
| $g$, m/s$^2$ | acceleration of gravity |
| $h$, m | maximal lake depth |
| $h_s$, m | lake surface deviation from horizontal |
| $h_{sed}$, m | the vertical size of sediments columns |
| $f$ | arbitrary water state variable (velocity component, temperature, salinity, gas concentration, etc.) |
| $k$, m$^2$/s$^2$ | turbulent kinetic energy (TKE) |
| $k_f$, m$^2$/s | turbulent diffusion/dissipation coefficient for variable $f$ |
| $L_x$, $L_y$, m | horizontal sizes of lake's horizontal cross-section $A(z)$ in $x$ and $y$ directions, respectively |
| $L_R$, m | Rossby deformation radius |
| $M$, s$^{-1}$ | shear frequency |
| $\mathbf{n}$ | an outer normal unit vector |





| | |
|---|---|
| $N$, s$^{-1}$ | Brunt-Väisälä frequency |
| $p$, Pa | in-water pressure |
| $p_a$, Pa | atmospheric pressure |
| Pr, n/d | Prandtl number |
| $R = 8.314$ J/(mol $*$ K) | universal gas constant |
| $R_f$ | sum of sources and sinks of variable $f$ |
| Ri, n/d | gradient Richardson number |
| $S$, m$^2$/s$^3$ | shear production of TKE |
| $S_{rad}$, $W/m^2$ | shortwave radiation flux in water, positive downwards |
| $t$, s | time |
| $T$, K | temperature |
| $T_{mp}$, K | melting point temperature |
| $\mathbf{u} = \{u_1, u_2, u_3\} = \{u, v, w\}$, m/s | 3D velocity vector in water |
| $T_{seiche}$, s | seiche period |
| $\mathbf{u_h} = \{u_1, u_2\} = \{u, v\}$, m/s | horizontal velocity vector in water |
| $\mathbf{u_a} = \{u_a, v_a\}$, m/s | wind speed vector |
| W, n/d | Wedderburn number |
| $\mathbf{x} = \{x_1, x_2, x_3\} = \{x, y, z\}$, m | 3D position vector |
| $Z$, m | vertical coordinate, originating at the bottom and pointing against gravity (used in the bubble model) |
| $z_{0b,eff}$, m | effective roughness length of a lake bottom |

## 8.2 Biogeochemistry

| | |
|---|---|
| $\alpha_i$, n/d | molar fraction of $i$-th gas in a bubble |
| $\alpha_{new} = 3$ m$^{-1}$ | a constant controlling the decrease of methane production with depth in sediments |
| $\alpha_{O_2,inhib} = 316.8$ m$^3$/mol | a constant controlling inhibition of methane production in sediments due to oxygen presence |
| $C_{x_s}$, mol/m$^3$ | bulk gas concentration in sediments, $x = CH_4, O_2$ |
| $E_{CH_4,s}$, mol/(m$^3 *$ s) | methane sink in sediments due to ebullition |
| $F_{B,i,k}(z)$, mol/(m$^2 *$ s) | bubble flux of $i$-th gas from $k$-th column of sediments at depth $z$ |
| $h_{bot}$, m | the depth of a point at the bottom where the bubble is released |
| $H_i$ | the Henry "constant" (temperature-dependent) of $i$-th gas |
| $k_{CH_4,s}$, m$^2$/s | molecular diffusivity of methane in sediments |
| $k_{CH_4,w}$, m$^2$/s | diffusion coefficient for methane in water |





| | |
|---|---|
| $k_{ge}$, m/s | gas exchange coefficient at the water-air interface ("piston velocity") |
| $k_{600}$, m/s | piston velocity at $Sc = 600$, $Sc$ – Schmidt number |
| $K_{CH_4,s} = 9.5 * 10^{-3}$ mol/m$^3$ | half-saturation constant in respect to $CH_4$ for methane oxidation in sediments |
| $K_{O_2,s} = 2.1 * 10^{-2}$ mol/m$^3$ | half-saturation constant in respect to $O_2$ for methane oxidation in sediments |
| $K_{CH_4,w} = 3.75 * 10^{-2}$ mol/m$^3$ | half-saturation constant in respect to $CH_4$ for methane oxidation in water |
| $K_{O_2,w} = 2.1 * 10^{-2}$ mol/m$^3$ | half-saturation constant in respect to $O_2$ for methane oxidation in water |
| $K_i$, m/s | $i$-th gas exchange coefficient in a bubble |
| $M_i$, mol | the content of $i$-th gas in a bubble |
| $n_b$, m$^{-3}$ | number density of bubbles in water |
| $n_g = 5$ | number of gases considered in a bubble |
| $O_{CH_4,s}$, mol/(m$^3$ * s) | aerobic methane oxidation rate in sediments |
| $P_{CH_4,s}$, mol/(m$^3$ * s) | production rate of methane in sediments |
| $P_0$, mol/(m$^3$ * s) | empirical constant, an amplitude of production rate of methane in sediments, $P_{CH_4,s}$ |
| $P_i$, Pa | $i$-th gas pressure in a bubble |
| $q_{10} = 2.3$, n/d | temperature dependence constant for methane production in sediments |
| $r_b$, m | bubble radius |
| $v_b$, m/s | bubble vertical velocity |
| $V_{max,s} = 1.11 * 10^{-5}$ mol/(m$^3$ * s) | methane oxidation potential in sediments |
| $V_{max,w} = 1.16 * 10^{-5}$ mol/(m$^3$ * s) | methane oxidation potential in water |
| $z_s$, m | depth in sediments, in respect to the sediments column' top |



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





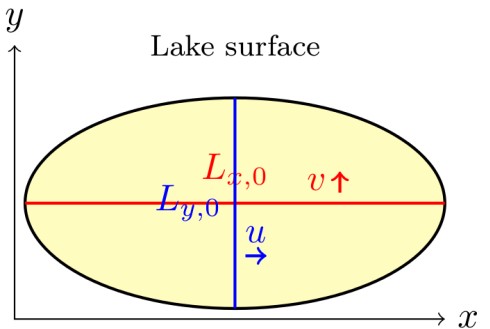
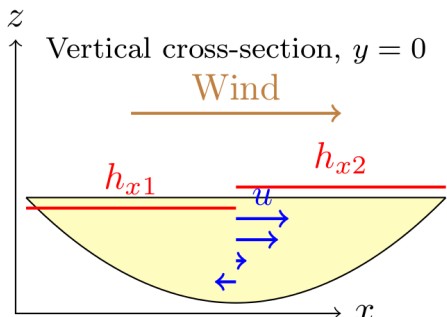

Figure 1: The sketch describing variables used in surface seiche parameterization. Lake surface is approximated by ellipse, whose axes are $L_{x0}$ and $L_{y0}$. Variable $h_{x1}$ is an average surface height of a left half domain of lake, $h_{x2}$ is that of a right half domain; $h_{y1}$ and $h_{y2}$ are defined analogously for the lower and upper halves.

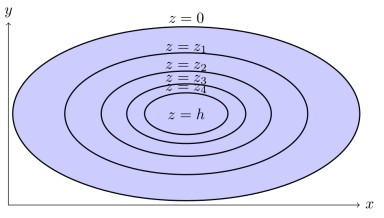

(a) The scheme for spatial sediments columns distribution. Here, horizontal cross-sections of sediments columns are confined by respective isobaths, i.e. $i$-th sediments column is bounded by $z_{i-1}$- and $z_i$-isobaths, $i = 1, ..., 4$. The bottom sediments column is of elliptic cross-section.

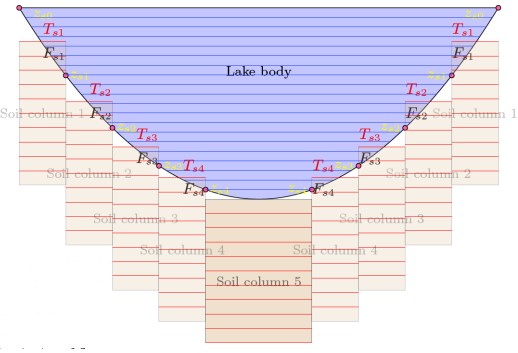

(b) Vertical cross-section of a water body and sediments columns in LAKE model, horizontal lines standing for computational levels.

Figure 2: Horizontal and vertical cross-sections of sediments columns in LAKE model.



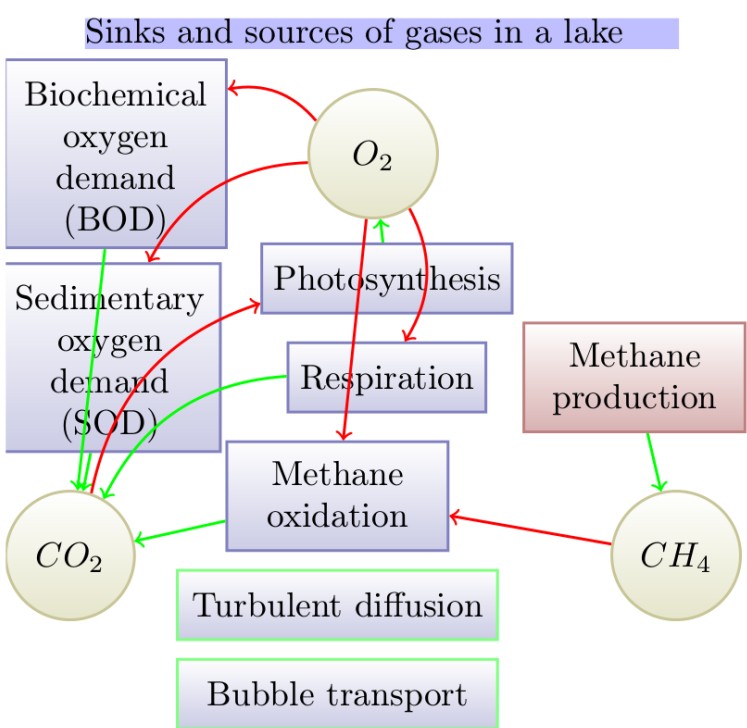

Figure 3: The $CH_4, CO_2, O_2$ storages and their interaction through biogeochemical processes in the model. Green arrows are sources, red arrows are sinks. Methane production is considered in sediments, other processes take place in water body

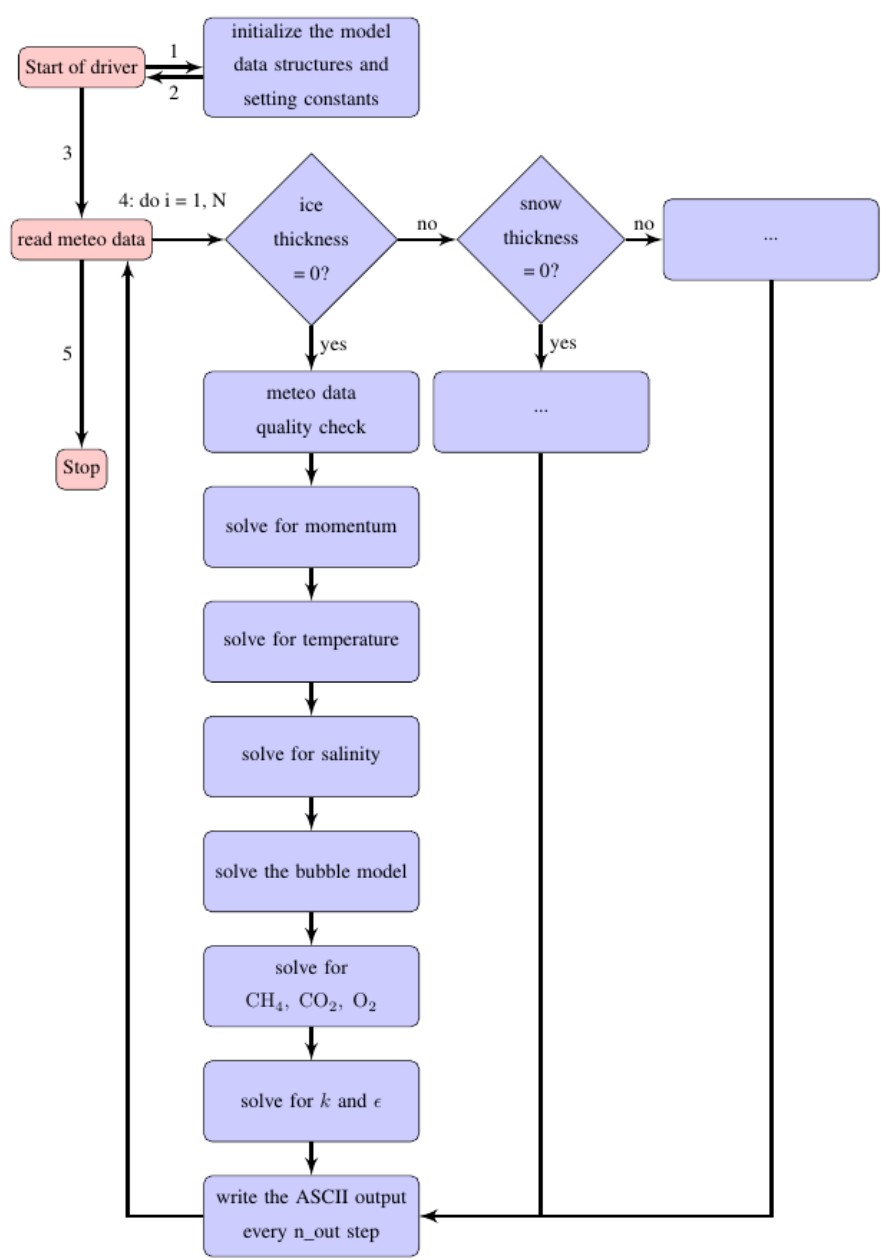

Figure 4: The flowchart of LAKE model. Pink boxes are operations of the driving program unit (may be an atmospheric/climate model). Blue boxes are operations of the model itself, N standing for the number of time steps, and n_out – for the period of output ( time steps). Each iteration of a cycle "do i=1, N" performs one time step of the model.

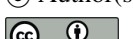


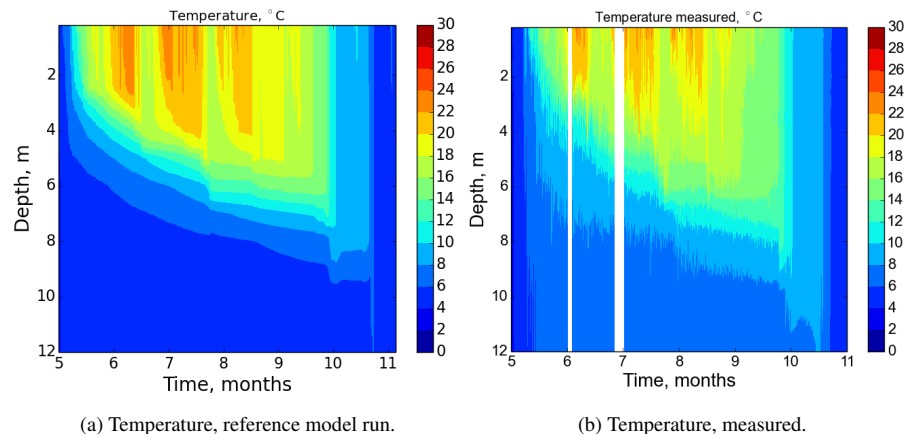

(a) Temperature, reference model run.       (b) Temperature, measured.

Figure 5: Time-depth distribution of temperature in Kuivajärvi Lake. Months at the horizontal axis are of 2013. White areas signify the absence of data

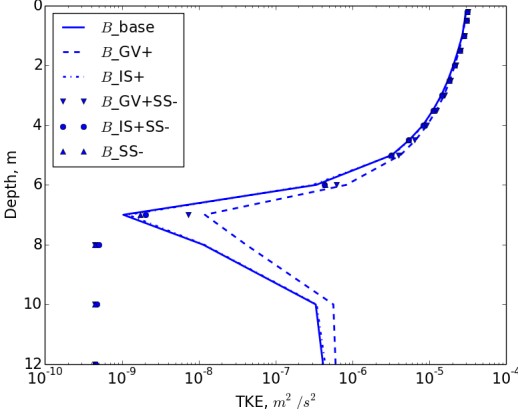

Figure 6: Mean TKE profile in Kuivajärvi Lake, July 2013, simulated. Model runs: "base" – baseline, "GV+" – including gravity waves shear parameterization (Gill, 1982; Mellor, 1989), "IS+" – including internal seiche mixing parameterization (Goudsmit, 2002), "GV+SS-" – the same as "GV+" but with surface seiches switched off, "IS+SS-" – the same as "IS+" but with surface seiches switched off, "SS-" – the same as "base" but with surface seiches switched off





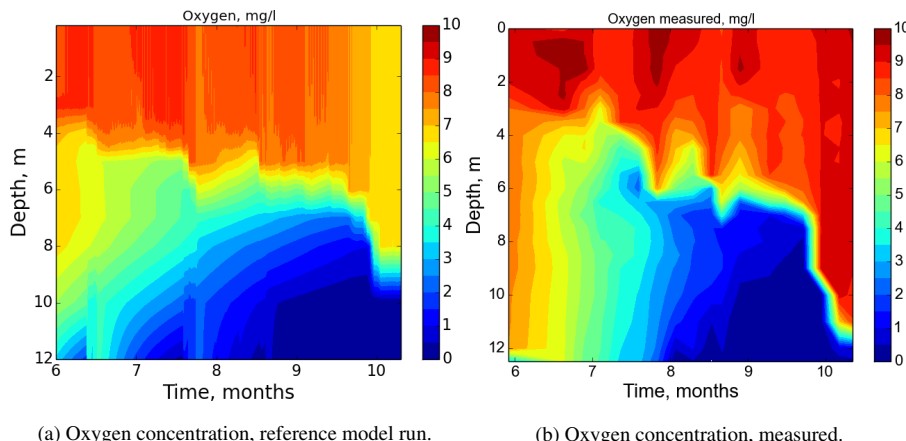

(a) Oxygen concentration, reference model run.    (b) Oxygen concentration, measured.

Figure 7: Time-depth distribution of dissolved oxygen in Kuivajärvi Lake. Months at the horizontal axis are of 2013.

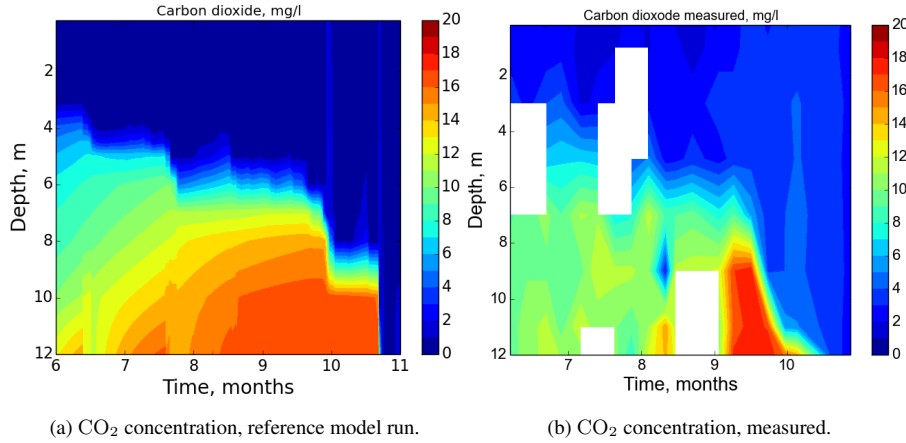

(a) $CO_2$ concentration, reference model run.    (b) $CO_2$ concentration, measured.

Figure 8: Time-depth distribution of dissolved carbon dioxide in Kuivajärvi Lake. Months at the horizontal axis are of 2013. White areas signify the absence of data.





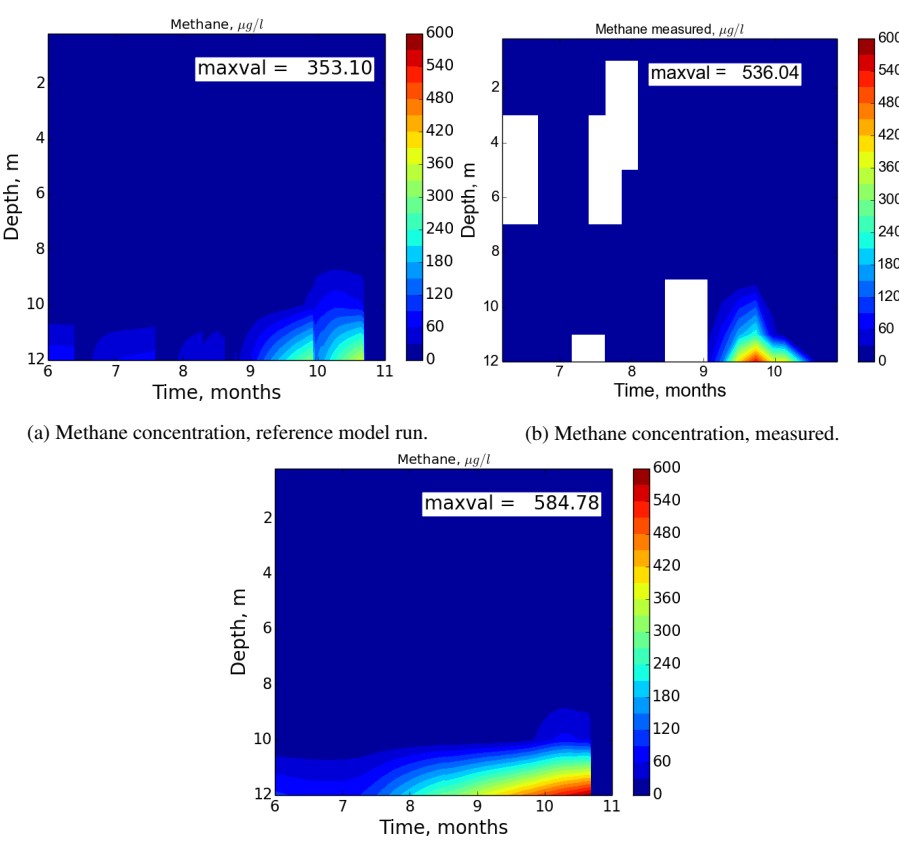

(a) Methane concentration, reference model run.  (b) Methane concentration, measured.

(c) Methane concentration, model run with surface se-

iches switched off (SS-).

Figure 9: Time-depth distribution of dissolved methane in Kuivajärvi Lake. Months at the horizontal

axis are of 2013. White areas signify the absence of data.



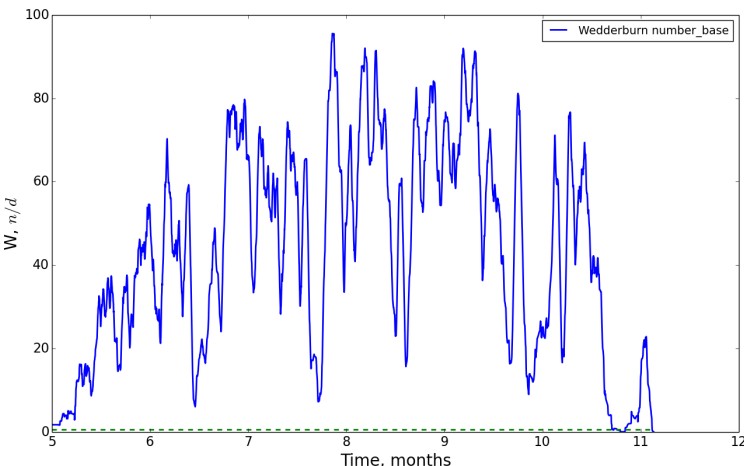

Figure 10: Time series of Wedderburn number from the model reference run. Months at the horizontal axis are of 2013. Dashed green line denotes critical value $W_{cr} = 0.5$

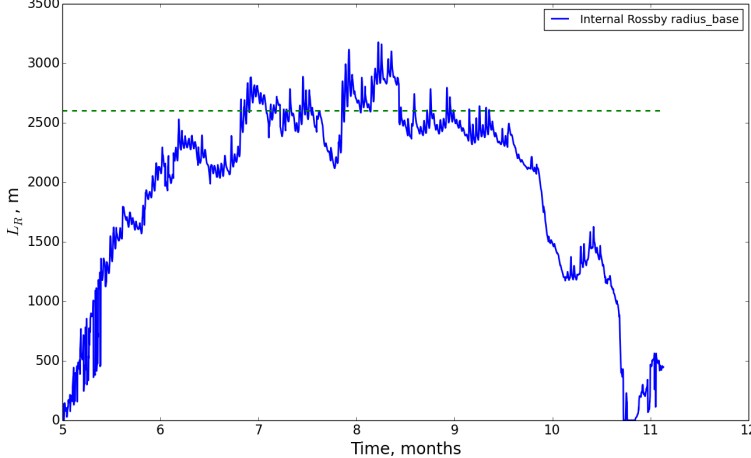

Figure 11: Time series of Rossby deformation radius from the model reference run. Months at the horizontal axis are of 2013. Dashed green line denotes the approximate Kuivajärvi Lake length, 2600 m





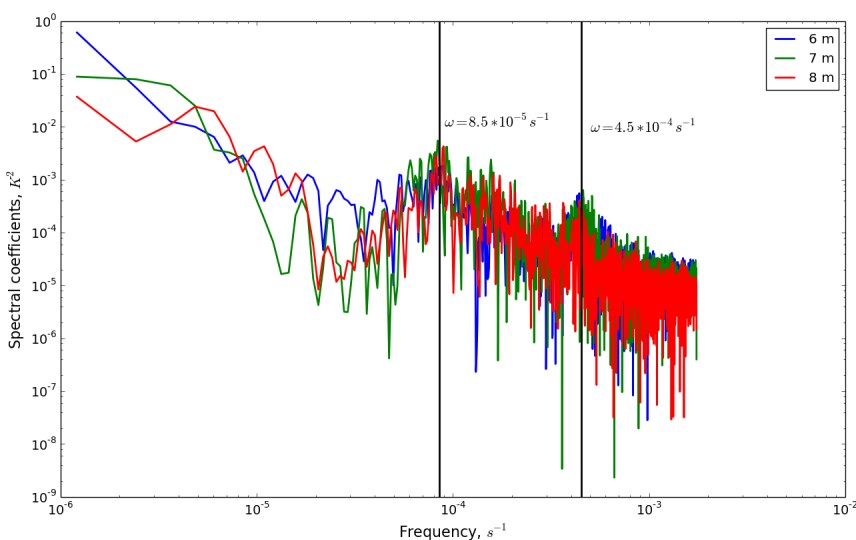

Figure 12: Fourier spectrum of water temperature fluctuations at depths 6 m, 7 m and 8 m. Two vertical lines point at maxima corresponding to $\omega = 8.5 * 10^{-5}$ s$^{-1}$ ($T_{seiche} \approx 20.5$ hour) and $\omega = 4.5 * 10^{-4}$ s$^{-1}$ ($T_{seiche} \approx 3.9$ hour)





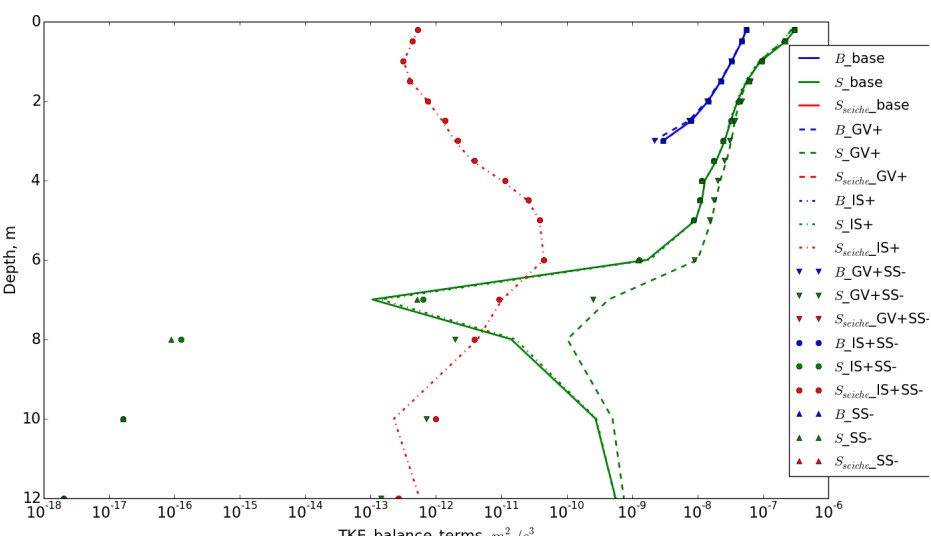

Figure 13: Mean TKE balance terms in Kuivajärvi Lake, July 2013, modeled. $B$ – production by buoyancy, $S$ – production by shear, $S_{seiche}$ – production by internal-seiche-induced shear. Model runs: "base" – baseline, "GV+" – including gravity waves shear parameterization (Gill, 1982; Mellor, 1989), "IS+" – including internal seiche mixing parameterization (Goudsmit, 2002), "GV+SS-" – the same as "GV+" but with surface seiches switched off, "IS+SS-" – the same as "IS+" but with surface seiches switched off, "SS-" – the same as "base" but with seiches switched off. Negative values of $B$ are not plotted, as well as zeros of $S_{seiche}$





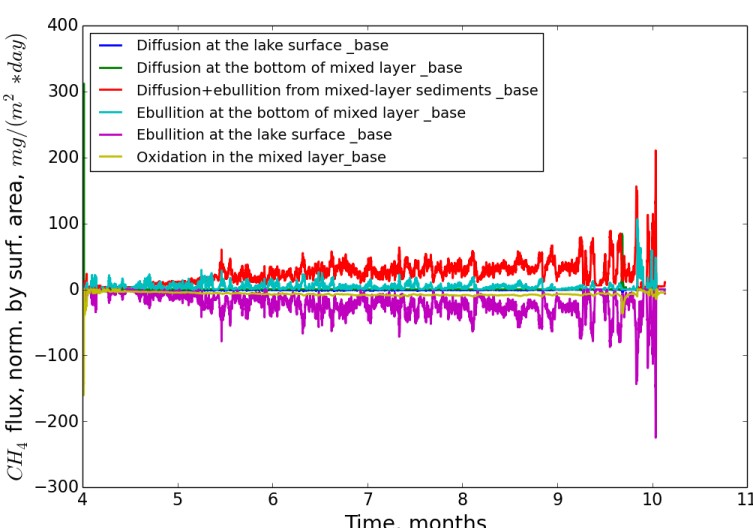

Figure 14: The components of methane balance in the surface mixed layer, normalized by lake surface area. Positive terms increase methane concentration in the mixed layer and negative ones are these decreasing $CH_4$ content. Suffix "_base" means baseline experiment





**Appendix A: Equation for horizontally averaged quantity in a lake**

Consider equation (1) and an auxiliary operator:

$$\tilde{f} = \int\limits_{A(z)} f \, dx \, dy. \tag{A1}$$

The cross-section of a lake with notations used in this derivation is given at Fig. **??**.

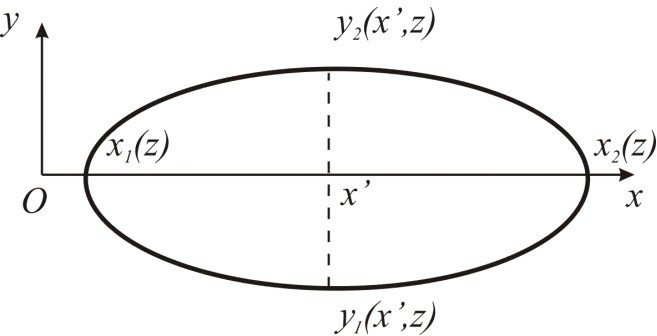

Figure 15: A lake horizontal cross-section

The integration operator (A1) possesses the following property:

$$\frac{\partial \tilde{f}}{\partial z} = \widetilde{\frac{\partial f}{\partial z}} + B_f, \tag{A2}$$

$$B_f = \int\limits_{x_1(z)}^{x_2(z)} \left[ \frac{\partial y_2}{\partial z} f(x, y_2, z) - \frac{\partial y_1}{\partial z} f(x, y_1, z) \right] dx, \tag{A3}$$

stemming from the Leibnitz integral rule. Now apply operator $\widetilde{(\ldots)}$ to (1), insert $\overline{f} = A\tilde{f}$, leading to

$$\frac{\partial A \overline{f}}{\partial t} = - \int\limits_{\Gamma_{A(z)}} f(\mathbf{u_h} \cdot \mathbf{n}) dl - \int\limits_{\Gamma_{A(z)}} (\mathbf{F_h} \cdot \mathbf{n}) dl - \frac{\partial A \overline{wf}}{\partial z} - \frac{\partial A \overline{F_z}}{\partial z} + B_{wf} + B_{F_z} + A\overline{R_f}, \tag{A4}$$

where we introduced $\mathbf{u_h} = \{u_1, u_2\}$, $\mathbf{F_h} = \{F_1, F_2\}$, and $\Gamma_{A(z)}$ is a boundary of $A$ at depth $z$. The first term to the right hand side of (A4) is a horizontal advection of property $f$ through boundaries of a water basin, i.e. the inflow from inlets, outflow by outlets and groundwater discharge. The second term represents non-advective horizontal fluxes at lake margins, whereas $B_*$ quantifies the effect of vertical fluxes at the lake bottom of depth $z$. Equation (A4) is the most general equation,





that is, however, difficult to implement without further simplifications. First, assume that the lake bottom is quasi-horizontal, and in this case the rigid boundary condition for velocity brings $w \approx 0$, $B_{wf} \approx 0$. Then, we suppose that $\mathbf{F} = \{F_1, F_2, F_3\}$ is normal to the bottom boundary, that is good approximation for diffusive transport, because it is proportional to a gradient of $f$, and this gradient is usually oriented almost perpendicular to the bottom surface. Therefore, $F_1 \approx 0$, $F_2 \approx 0$, vanishing the second term to the right hand side of (A4). We also can decompose the vertical advection as $\overline{wf} = \overline{w}\,\overline{f} + \overline{w'f'}$, $w' = w - \overline{w}$, $f' = f - \overline{f}$. After these modifications, (A4) devolves to:

$$\frac{\partial A\overline{f}}{\partial t} = -\int\limits_{\Gamma_{A(z)}} f(\mathbf{u_h} \cdot \mathbf{n})dl - \frac{\partial A\overline{w}\,\overline{f}}{\partial z} - \frac{\partial A\overline{w'f'}}{\partial z} - \frac{\partial A\overline{F_z}}{\partial z} + B_{F_z} + A\overline{R_f}. \tag{A5}$$

At this stage it is timely to distinguish between turbulent and non-turbulent fluxes, namely $F_z = F_{tz} + F_{nz}$, and define "effective" turbulent flux, $\overline{F_{tz}^*} = \overline{F_{tz}} + \overline{w'f'}$. This effective turbulent flux includes horizontally-averaged small-scale turbulent flux $(\overline{F_{tz}})$ and the flux mediated by large-scale flow structures, $\overline{w'f'}$. We also assume that the total non-advective flux $F_z$ at the bottom is the same at all bottom locations of the depth $z$, i.e. $\forall z:\ F_z(x,y) = \mathrm{const}, (x,y) \in \Gamma_{A(z)}$. Then, taking into account the above hypotheses and

$$B_1 = \int\limits_{x_1(z)}^{x_2(z)} \left[ \frac{\partial y_2}{\partial z} - \frac{\partial y_1}{\partial z} \right] dx = \frac{dA}{dz}, \tag{A6}$$

we transform (A5) to

$$\frac{\partial A\overline{f}}{\partial t} = -\int\limits_{\Gamma_{A(z)}} f(\mathbf{u_h} \cdot \mathbf{n})dl - \frac{\partial A\overline{w}\,\overline{f}}{\partial z} - \frac{\partial A\overline{F_{nz}}}{\partial z} - \frac{\partial A\overline{F_{tz}^*}}{\partial z} + \frac{dA}{dz}(F_{nz,b}(z) + F_{tz,b}(z)) + A\overline{R_f}, \tag{A7}$$

where $F_{*,b}(z)$ denote bottom values of fluxes at depth $z$. The mean vertical velocity, $w$, may be expressed from the horizontally integrated continuity equation (2):

$$\frac{\partial A\overline{w}}{\partial z} = B_w - \int\limits_{\Gamma_{A(z)}} (\mathbf{u_h} \cdot \mathbf{n})dl, \tag{A8}$$

where $B_w \approx 0$ according to assumption of quasi-horizontal bottom. This means, $w$ arises from disbalance between inflows and outflows and subsequent water level change. For the LAKE model hasn't been applied for water bodies with significant water level change, the term with $w$ is omitted in (A9) in the model equation set. In order (A9) equation to become tractable we use the following assumptions:

- the 'effective' turbulent flux may be represented via the gradient of mean quantity: $\overline{F_{tz}^*} = -k_f \frac{\partial \overline{f}}{\partial z}$;



– the source averaged horizontally, $\overline{R_f(f,...)}$, may be approximated as the same function of mean values, $\overline{R_f(f,...)} = R_f(\overline{f},...)$.

Substituting these statements into (A9), we finally get:

$$\frac{\partial \overline{f}}{\partial t} = -\frac{1}{A}\int\limits_{\Gamma_{A(z)}} f(\mathbf{u_h} \cdot \mathbf{n})dl + \frac{1}{A}\frac{\partial}{\partial z}\left(Ak_f\frac{\partial \overline{f}}{\partial z}\right) - \frac{1}{A}\frac{\partial A\overline{F_{nz}}}{\partial z} + \frac{1}{A}\frac{dA}{dz}[F_{nz,b}(z)+F_{tz,b}(z)] + R_f(\overline{f},...).$$

$$(A9)$$

## Appendix B: Standard $k - \epsilon$ model

The prognostic equations for TKE, $k$ and its dissipation rate, $\epsilon$, take the form:

$$\frac{\partial k}{\partial t} = \frac{1}{A}\frac{\partial}{\partial z}A\left(\nu_m + \frac{\nu}{\sigma_k}\right)\frac{\partial k}{\partial z} + S + B - \epsilon, \tag{B1}$$

$$\frac{\partial \epsilon}{\partial t} = \frac{1}{A}\frac{\partial}{\partial z}A\left(\nu_m + \frac{\nu}{\sigma_\epsilon}\right)\frac{\partial \epsilon}{\partial z} + \frac{\epsilon}{k}\left(c_{\epsilon 1}S + c_{\epsilon 3}B - c_{\epsilon 2}\epsilon\right), \tag{B2}$$

$$S = \nu\left[\left(\frac{\partial u}{\partial z}\right)^2 + \left(\frac{\partial v}{\partial z}\right)^2\right], \tag{B3}$$

$$B = -\frac{g}{\rho_{w0}}\nu_T\left(\alpha_T\frac{\partial T}{\partial z} + \alpha_s\frac{\partial s}{\partial z}\right), \tag{B4}$$

$$\nu = C_e\frac{k^2}{\epsilon}, \tag{B5}$$

$$\nu_T = C_{e,T}\frac{k^2}{\epsilon}. \tag{B6}$$

Here, $\alpha_T(T,s)$ designates thermal expansion coefficient, $\alpha_s(T,s)$ - expansion coefficient in respect

to salinity, $s$. The coefficients and stability functions of the model are given in Table A1.

Boundary conditions are the same at upper and lower boundaries, and exact for logarithmic boundary layer (Burchard and Petersen, 1999):

$$\frac{\nu}{\sigma_k}\frac{\partial k}{\partial z}\bigg|_{z=0,h} = 0,$$

$$\frac{\nu}{\sigma_\epsilon}\frac{\partial \epsilon}{\partial z}\bigg|_{z=0,h} = -C_{e,0}^{3/4}\frac{\nu}{\sigma_\epsilon}\frac{k^{3/2}}{\kappa z_0^2},$$

where $C_{e,0} = 0.09$ designates a reference value for momentum stability function, $z_0 = 10^{-2}$ m – an empirical parameter, $\kappa = 0.38$ – von Karman constant.





Table A1: Coefficients of standard $k - \epsilon$ model

| Constants | |
|---|---|
| $\sigma_k$ | 1 |
| $\sigma_\epsilon$ | 1.111 |
| $c_{\epsilon 1}$ | 1.44 |
| $c_{\epsilon 2}$ | 1.92 |
| $c_{\epsilon 3}$ | 1.14 if $B > 0$, -0.4 otherwise |

| Stability functions | |
|---|---|
| $C_e$ | Stability function for momentum (Canuto et al., 2001) |
| $C_{e,T}$ | Stability function for scalars (Canuto et al., 2001) |

**Appendix C: Calibration of horizontal pressure gradient parameterization**

Consider fluctuations of surface level and a velocity of the flow that are homogeneous in $y$, developing in a channel of parallelepiped form, with depth $h$ and horizontal dimensions $L_x$ and $L_y$, neglecting friction and rotational effects. Under these conditions, momentum and mass conservation in a 1D approximation takes the form

$$\frac{\partial u}{\partial t} = -g \frac{\partial h}{\partial x}, \tag{C1}$$

$$g \frac{\partial h}{\partial x} = g \frac{h_1 - h_0}{\alpha L_x}, \tag{C2}$$

$$\frac{\partial h_1}{\partial t} = -\frac{\partial h_0}{\partial t} = 2A^{-1} \overline{u}^{yz} h L_y = 2\overline{u}^{yz} h L_x^{-1}, \tag{C3}$$

where $\alpha$ - a constant to be defined later, the operator $\overline{f}^{yz}$ averages the quantity $f$ in a plane $x = const$, $A = L_x L_y$ is a horizontal cross-section area of a channel, $h_0$ is an average surface level over a "left" part of the channel $[0, L_x/2] \times [0, L_y]$, and $h_1$ is that for the right part, $[L_x/2, L_x] \times [0, L_y]$. Approximation (C2) means that we confine ourselves to reproducing 1-st horizontal seiche mode, which is, however, often reported as the most prominent on lakes (Hutter et al., 2011). From (C1) we get

$$\frac{\partial \overline{u}^{yz}}{\partial t} = -g \frac{\partial h}{\partial x}. \tag{C4}$$





Using (C4), (C2) and (C3) yields:

$$\frac{\partial^2 \overline{u}^{yz}}{\partial t^2} = -\frac{\partial}{\partial t}\left(g\frac{\partial h}{\partial x}\right) = -\frac{g}{\alpha L_x}\frac{\partial}{\partial t}(h_1 - h_0) = -\frac{4g\overline{u}^{yz}h}{\alpha L_x^2}. \tag{C5}$$

Substituting here $\overline{u}^{yz} \sim \exp(-i\omega t)$, we get formulas for the frequency and period of surface seiche:

$$\omega = \frac{2\sqrt{gh}}{\sqrt{\alpha}L_x}, \; T = \frac{\pi\sqrt{\alpha}L_x}{\sqrt{gh}}, \tag{C6}$$

and then, comparing to a Merian formula (Merian, 1828)

$$T = \frac{2L_x}{\sqrt{gh}}, \tag{C7}$$

gives

$$\alpha = \frac{4}{\pi^2} \approx 0.41, \tag{C8}$$

so that the value of $\alpha$ ensuring correct period of the 1-st horizontal seiche mode significantly differs from a "natural" choice $\alpha = 0.5$.

In the case of motions in both $x$ and $y$ directions, the formula, analogous to (C5) is valid for $y$-component of velocity, $v$. Equations for $u$ and $v$ are decoupled in this approximation, so that fluctuations of $u$ and $v$ develop independently. This is different from what we have in shallow water equations where $u$ and $v$ are coupled via divergence in mass continuity equation and corresponding surface elevation change. Hence, the 1-st mode seiche model described above is yet to be generalized to a 2D case in a way to include horizontal divergence. Still, for lacustrine environment applications, our approximation allows to generate TKE below thermocline that is principally unachievable in standard $k - \epsilon$ model.

## Appendix D: Stationary Richardson number for $k - \epsilon$ model with Goudsmit seiche parameterization

An extension of standard $k - \epsilon$ model was proposed in (Goudsmit, 2002) to introduce additional TKE production by shear induced by internal seiches. The corresponding extra term, $S_{seiche}$, has been added to production by mean vertical shear:

$$S = \nu M^2 + S_{seiche}, \tag{D1}$$

$$S_{seiche} = -\frac{1 - C_{diss}\sqrt{C_{d,bot}}}{\rho_{w0}cA_b}\gamma\frac{1}{A}\frac{dA}{dz}N^{2q}E_{seiche}^{3/2}, \tag{D2}$$



where $C_{d,bot} \approx 0.002$ is the bottom drag coefficient, $A_b$ - the total bottom area, $c$ - normaliz-
ing constant, $\gamma$ - coefficient, characterizing dissipation of seiche energy, $E_{seiche}$. The combination
$C_{diss}\sqrt{C_{d,bot}} \approx 0.4$ $(C_{diss} = 10)$ is a fraction of seiche energy, trasnferred to heat in a visous bot-
tom sublayer. Hereafter, we will assume $q = 1$ for simplicity. In (Goudsmit, 2002), this was a cali-
bration parameter, taking values close to unity. From stationarity condition in seiche energy equation
(equation (15) in (Goudsmit, 2002)) we have a balance between energy transferred from wind drag
work on a lake surface, and seiche dissipation:

$$\alpha A_0 \rho_a C_d (u_a^2 + v_a^2)^{3/2} = \gamma E_{seiche}^{3/2}. \tag{D3}$$

Here, $\alpha \approx 2 \cdot 10^{-3}$. Now we will use $k - \epsilon$ model equations (B1) and (B2) under stationarity and
homogeneity conditions:

$$S + B - \epsilon = 0, \tag{D4}$$

$$c_{\epsilon 1} S + c_{\epsilon 3} B - c_{\epsilon 2} \epsilon = 0. \tag{D5}$$

Substituting (D3) into (D2), and then (D1) to (D4) and (D5), eliminating $\epsilon$ from the latter two, we
get:

$$-C_e \frac{k^2}{\epsilon} \Delta c_{\epsilon 21} + C_{e,T} \frac{k^2}{\epsilon} \Delta c_{\epsilon 23} \mathrm{Ri}_{st} - \Delta c_{\epsilon 21} C_s \mathrm{Ri}_{st} (u_a^2 + v_a^2)^{3/2} = 0, \tag{D6}$$

where we defined a new value $C_s = -\frac{(1 - C_{diss}\sqrt{C_{d,bot}}) A_0 \rho_a C_d \alpha}{\rho_{w0} c A_b A} \frac{dA}{dz} > 0$, that is a constant in time for
a given lake. Then, assume that $C_e \frac{k^2}{\epsilon} \to \nu_0$ if $\mathrm{Ri} \to \mathrm{Ri}_{st}$. The parameter $\nu_0$ is of arbitrary choice,
however, we presume it to be a small value (e.g. $\nu_0 \sim \nu_m$), since it is eddy diffusivity on the edge of
regime of decaying homogeneous turbulence. Thence,

$$\mathrm{Ri}_{st} = \frac{\mathrm{Pr}\Delta c_{\epsilon 21}}{\Delta c_{\epsilon 23} - \nu_0^{-1} \mathrm{Pr} C_s \Delta c_{\epsilon 21} (u^2 + v^2)^{3/2}}. \tag{D7}$$

In the original work (Goudsmit, 2002) parameter $\alpha$ was calibrated to be $\sim 6 * 10^{-3}$. After substituting
typical values of parameters mentioned above and morphometry data of Kuivajärvi Lake, we got
$\mathrm{Ri}_{st} = 0.30$.