# Peer review of "LAKE 2.0: a model for temperature, methane, carbon dioxide and oxygen dynamics in lakes"

_Geoscientific Model Development, 2015_

## Referee Comment (RC1) · Anonymous Referee #1 · 11 Mar 2016

This paper presents 1D model for a lake, describing an evolution of horizontally averaged vertical distribution of water temperature, momentum, concentrations of oxygen, carbon dioxide and methane. It includes a module simulating heat and moisture processes in sediments, a module for a gas bubbles, and biogeochemistry modules for CO2, CH4 and O2. The whole set of coupling modules, which makes LAKE 2.0 model, was tested against the Kuivajarvi Lake observations and showed a fair agreement. The paper presents a substantial advance in modeling of geophysical enclosed water bodies. The methods and assumptions are valid and clearly outlined. The interpretations and concentrations are properly supported by results.

The abstract does not provide a complete summary, but could be revised to include more results.

[Figure]

English is not my native language. Some sentences sound strange to me, but I cannot give an expertise.

I found the paper to be very interesting and well-written. The experimental set-up is well-conceived, and the analysis seems sound and comprehensive. The model simulations and configurations are well documented. I have no hesitation in recommending this manuscript to be published. I recommend to improve quality of plots, some of them look bad. I do have a couple of minor comments that the authors can address at their convenience.

3-5. I would recommend to omit this sentence or to rephrase, because: a) it is not a good idea to make reader to evaluate author's knowledge from the very beginning (from abstract), b) the "knowledge" is changing and future readers won't understand what it is about. If authors want to rephrase I would suggest to indicate what is exactly included in their general form of 1D diffusion-type equation.

8-9. It is a good result, but not the only one and not the best one. I would suggest the authors to extend the list.

25. I would recommend a brief outline of the models listed here, showing what are their advantages and disadvantages, what ideas were taken and what approaches were used and developed in presented model.

45-46. It is not clear what does it mean. Please, omit or rephrase.

50-52. What kind of problems this model is supposed to solve? I think this is the key question in designing a model, but it is not answered here.

54-55. Why vertical turbulent flux through hypolimnion and metalimnion are of special concern? Please, explain.

55-56. What is going to be a development? What was wrong with LAKE? Please, describe a progress.

68. Please, give more details about "certain physical processes" here.

90. "c" cannot be a specific heat because of conflict of units.

99. A(z) is an area (!) of horizontal cross-section, isn't it? Not a cross-section.

99. Why diffusion and dissipation are slashed here. They are quite different processes. Kf is not a diffusion (neither a dissipation) but is used to parameterize diffusion, but parameterization of dissipation could not necessarily use it.

169-170. The above conditions (166-168) say nothing about gas concentrations, how could gas concentrations be affected by them and what are the conditions for gas concentrations?

284-286. Some sentences, like this one, attribute a model description to a specific lake study, but the aim of the paper is a model development. I would suggest to address the absence of methane production in model to a further development not to "the lake under study".

379-380. I was confused with the mixture of variables and their units here. What if to specify units somewhere else? They are all listed in "List of symbols" along with their units. Why not to omit units in text?

383-384. It is not clear for me why argon is important when consider the bubble gases and water vapor is not. Could you explain, please?

600-620. Could you explain somehow the saw-shape methane concentration at the bottom in "reference" model results (Fig 9a)?

810. Good place to introduce BOD and SOD abbreviations, because of their further use.

861. "Wee" .

903. (and somewhere else) Change "U.Svensson" to "Svensson" or add initials to

others, for example, "G.L. Mellor and G.-H. Goudsmit" (914)

Figures 5-14. Poor quality of lines and text. I only have an idea what it is written in legends after multiple zoom.

Figure 13. Some lines are declared in legend but not available in plot.

---

## Referee Comment (RC2) · Anonymous Referee #2 · 15 Mar 2016

This paper presents excellent description of the latest version of 1D lake model "LAKE". The model solves horizontally averaged equations for the heat, gases and momentum transport within water body. General description for main processes represented in the model is provided. The model is validated against data gathered at lake Kuivajärvi (Southern Finland). Basic results are well illustrated. The model satisfactorily reproduces observed patterns of seasonal variations of temperature and gases in lake Kuivajärvi with calibration of only two constants relevant to CH4 production and consumption. The authors emphasize the role of the internal oscillations and the thermocline turbulence in vertical transfer of dissolved.

The paper is well written and well structured. I recommend the paper for publication in its present form.

---

## Author Comment (AC1)

The authors are grateful to the anonymous referee for carefully reading the manuscript and proposing a number of valuable improvements to the text. They are addressed in the table below.

| Referee's comment                                                                                                                                                                                                                                                                                                                                                                                                                              | Authors' response                                                                                                                                                                                                                                                                                                                                                                                                                                                                                                                                                                                                                                                                                                                                                                                                                                                                 |
|------------------------------------------------------------------------------------------------------------------------------------------------------------------------------------------------------------------------------------------------------------------------------------------------------------------------------------------------------------------------------------------------------------------------------------------------|-----------------------------------------------------------------------------------------------------------------------------------------------------------------------------------------------------------------------------------------------------------------------------------------------------------------------------------------------------------------------------------------------------------------------------------------------------------------------------------------------------------------------------------------------------------------------------------------------------------------------------------------------------------------------------------------------------------------------------------------------------------------------------------------------------------------------------------------------------------------------------------|
| 3-5. I would recommend to omit this
sentence or to rephrase, because: a) it is
not a good idea to make reader to
evaluate author's knowledge from the
very beginning (from abstract), b) the
"knowledge" is changing and future
readers won't understand what it is about.
If authors want to rephrase I would
suggest to indicate what is exactly
included in their general form of 1D
diffusion-type equation. | The sentence is removed.                                                                                                                                                                                                                                                                                                                                                                                                                                                                                                                                                                                                                                                                                                                                                                                                                                                          |
| 8-9. It is a good result, but not the only
one and not the best one. I would suggest
theauthors to extend the list.                                                                                                                                                                                                                                                                                                                      | The abstract is extended by the following text:
"The model is validated vs. comprehensive
observational dataset gathered at Kuivajarvi
Lake (Southern Finland) demonstrating a fair
agreement. The value of a key calibration
constant, regulating the magnitude of methane
production in sediments, corresponded well to
that obtained from other two lakes. We
demonstrated via surface seiche
parameterization that the near-bottom turbulence
induced by surface seiches is likely to
significantly affect CH 4 accumulation there.
Furthermore, our results suggest that a gas
transfer through thermocline under intense
internal seiche motions is a bottleneck in
quantifying greenhouse gas dynamics in dimictic
lakes, calling for further research."                                                        |
| 25. I would recommend a brief outline of
the models listed here, showing what are
their advantages and disadvantages, what
ideas were taken and what approaches
were used and developed in presented
model.                                                                                                                                                                                                                     | This paragraph is rewritten as follows
(amendments denoted by bold ):
"Concomitantly with growing awareness of
lakes significance for current and future climate
change, few attempts have been made to develop
lake models, incorporating thermodynamics,
turbulence and biogeochemistry in order to
simulate $CH_4$ and $CO_2$ in natural water bodies
(Stepanenko et al., 2011;
Kessler et al., 2012; Tan et al., 2015).
study the response of lakes and their greenhouse
gas emissions to the future climate change \
(Tan and Zhuang, 2015b) and to assess the
relevant feedbacks through implementation of
biogeochemical lake models into the Earth
system models. These lake models rely on
well-established 1D thermodynamic and
turbulence closure schemes, whereas |

[revised manuscript text omitted]

|                                                                                                                                                                                                                                                               | column and fluxes to the atmosphere, and (iii)
vertical transport of water properties in order
to ensure (i) and (ii). Vertical turbulent flux of
dissolved gases through hypolimnion and
metalimnion are of special concern in this work,
since CO 2 and CH 4 mostly originate in the
hypolimnion, while the major interest for
community is how much of these species
evade to the atmosphere. The lake model,
developed here is based on LAKE model, that
has been continuously advanced during last
decade in Moscow State University
(Stepanenko and Lykossov, 2005; Stepanenko
et al., 2011) and was extensively validated in
LakeMIP experiments (Stepanenko et al.,
2010, 2013, 2014) in terms of lake temperature
and energy fluxes. The main development of
LAKE 2.0 compared to LAKE includes
biogechemical module, describing processes
related to O 2 , CO 2 and CH 4 dynamics,
multiple columns of sediments (facilitating
heat and gas exchange between water column
and sediments at different depths) and
surface seiche parameterization." |
|---------------------------------------------------------------------------------------------------------------------------------------------------------------------------------------------------------------------------------------------------------------|------------------------------------------------------------------------------------------------------------------------------------------------------------------------------------------------------------------------------------------------------------------------------------------------------------------------------------------------------------------------------------------------------------------------------------------------------------------------------------------------------------------------------------------------------------------------------------------------------------------------------------------------------------------------------------------------------------------------------------------------------------------------------------------------------------------------------------------------------------------------------------------------------------------------------------------------------------------------------------------------------------------------------------------------------------------------------------------------------------------------------------------------------------------------------------------|
| 68. Please, give more details about
"certain physical processes" here.                                                                                                                                                                                     | The new variant of the sentence:
"In Section 5, we analyze results of reference
experiment as well as of sensitivity experiments,
elucidating significance of vertical gas transport
induced by surface and internal seiches. "                                                                                                                                                                                                                                                                                                                                                                                                                                                                                                                                                                                                                                                                                                                                                                                                                                                                                                                                       |
| 90. "c" cannot be a specific heat because of conflict of units.                                                                                                                                                                                               | Equation (1) contained a typo, now it is
corrected:
$c\frac{\partial f}{\partial t} = -c\frac{\partial u_i f}{\partial x_i} - \frac{\partial F_i}{\partial x_i} + R_f(f,)$                                                                                                                                                                                                                                                                                                                                                                                                                                                                                                                                                                                                                                                                                                                                                                                                                                                                                                                                                                                                         |
| 99. A(z) is an area (!) of horizontal cross-
section, isn't it? Not a cross-section.                                                                                                                                                                       | Yes, corrected.                                                                                                                                                                                                                                                                                                                                                                                                                                                                                                                                                                                                                                                                                                                                                                                                                                                                                                                                                                                                                                                                                                                                                                          |
| 99. Why diffusion and dissipation are
slashed here. They are quite different
processes. Kf is not a diffusion (neither a
dissipation) but is used to parameterize
diffusion, but parameterization of
dissipation could not necessarily use it. | Agree, this textblock is rewritten as follows:
"k f the turbulent diffusivity (conductivity for
temperature, viscosity for momentum)
coefficient for variable f"                                                                                                                                                                                                                                                                                                                                                                                                                                                                                                                                                                                                                                                                                                                                                                                                                                                                                                                                                                                                     |
| 169-170. The above conditions (166-168)
say nothing about gas concentrations,
how could gas concentrations be affected
by them and what are the conditions for
gas concentrations?                                                                | The paragraph is amended (changes marked by
bold):
"The same options hold for CH₄
concentration, as diffusion-type equations are
solved in the water column and in each                                                                                                                                                                                                                                                                                                                                                                                                                                                                                                                                                                                                                                                                                                                                                                                                                                                                                                                                                                                                      |

|                                                                                                                                                                                                                                                                                                     | sediment column for this property as well (see
Sections 2.6.1 and 2.6.2). We found that the
first option provides reasonable results for
temperature and especially for CH 4
concentrations (see below in the paper), whereas
the second one needs calibration of parameters
entering the flux-gradient relationship in the
bottom boundary layer. The marginal heat flux is
calculated using the same temperature ( CH 4
concentration) and flux continuity condition,
that is facilitated by the solution of vertical heat
(CH 4 ) transfer in sediments below sloping
bottom (see details in Section 2.5)."                                                   |
|-----------------------------------------------------------------------------------------------------------------------------------------------------------------------------------------------------------------------------------------------------------------------------------------------------|---------------------------------------------------------------------------------------------------------------------------------------------------------------------------------------------------------------------------------------------------------------------------------------------------------------------------------------------------------------------------------------------------------------------------------------------------------------------------------------------------------------------------------------------------------------------------------------------------------------------------------------------------------------------------------------------------------------------------------------|
| 284-286. Some sentences, like this one,
attribute a model description to a specific
lake study, but the aim of the paper is a
model development. I would suggest to
address the absence of methane
production in model to a further
development not to "the lake under
study". | The sentence is corrected to:
"Deep CH 4 production from old organics near
the bottom of talik is included in the model
(Stepanenko et al., 2011), but in Kuivajarvi Lake
simulation presented here it is switched off
because this lake is not a thermokarst one."                                                                                                                                                                                                                                                                                                                                                                                                                                         |
| 379-380. I was confused with the mixture
of variables and their units here. What if
to specify units somewhere else? They are
all listed in "List of symbols" along with
their units. Why not to omit units in text?                                                                    | All units in the text are omitted, except for where values of variables are given.                                                                                                                                                                                                                                                                                                                                                                                                                                                                                                                                                                                                                                                    |
| 383-384. It is not clear for me why argon
is important when consider the bubble
gases and water vapor is not. Could you
explain, please?                                                                                                                                                   | The paragraph in the corrected text ( bold denoting changes ):
"Five gases are considered in a bubble: CH 4 ,
CO 2 , O 2 , N 2 and Ar. Water vapour constitutes
minor contribution to a bubble pressure, and
therefore neglected. Indeed, the saturated vapour
pressure at 20 C is 23.4 hPA that is 2% of
atmospheric pressure. This is the upper estimate
for water vapour pressure contribution in
bubbles, as the pressure increases with depth,
and saturation vapour pressure decreases, due
to water temperature drop. Similar estimates
hold for Ar, though it is formally included in
the bubble model. " |
| 600-620. Could you explain somehow the
saw-shape methane concentration at the
bottom in "reference" model results (Fig
9a)?                                                                                                                                                                | The new paragraph is added here:
"The "SS-" experiment (Fig. 10) provides a clue
for explanation of the saw-like pattern of CH 4
concentration in the reference model run (Fig.
9a). The closer joint inspection of Figs. 7a and
9a reveals that CH 4 drops near bottom coincide
in time with O 2 jumps. Oxygen jumps are
evidently caused by enhanced vertical mixing, as
there are no oxygen sources at large depths. In                                                                                                                                                                                                                                            |

|                                                                                                                                                   | contrast, such mixing events completely absent
when surface seiches are switched off (Fig. 10).
This leads us to a firm conclusion, that the
variability of mixing and respective gases
concentrations variations are caused by surface
seiches intensified by increased wind forcing
events." |
|---------------------------------------------------------------------------------------------------------------------------------------------------|------------------------------------------------------------------------------------------------------------------------------------------------------------------------------------------------------------------------------------------------------------------------------------------------------------------|
| 810. Good place to introduce BOD and SOD abbreviations, because of their further use.                                                             | Agree, done.                                                                                                                                                                                                                                                                                                     |
| 861. "Wee".                                                                                                                                       | Corrected.                                                                                                                                                                                                                                                                                                       |
| 903. (and somewhere else) Change
"U.Svensson" to "Svensson" or add initials
to others, for example, "G.L. Mellor and
GH. Goudsmit" (914) | Agree, initials are removed everywhere.                                                                                                                                                                                                                                                                          |
| Figures 5-14. Poor quality of lines and text. I only have an idea what it is written in legends after multiple zoom.                              | All the figures are enlarged.                                                                                                                                                                                                                                                                                    |
| Figure 13. Some lines are declared in legend but not available in plot.                                                                           | The legend is corrected accordingly.                                                                                                                                                                                                                                                                             |